# AlphaZeroES: Direct Score Maximization Can Outperform Planning Loss Minimization in Single-Agent Settings

## Abstract

Planning at execution time has been shown to dramatically improve performance for AI agents. A well-known family of approaches to planning at execution time in single-agent settings and two-player zero-sum games are AlphaZero and its variants, which use Monte Carlo tree search together with a neural network that guides the search by predicting state values and action probabilities. AlphaZero trains these networks by minimizing a planning loss that makes the value prediction match the episode return, and the policy prediction at the root of the search tree match the output of the full tree expansion. AlphaZero has been applied to various single-agent environments that require careful planning, with great success. In this paper, we explore an intriguing question: in single-agent settings, can we outperform AlphaZero by directly maximizing the episode score instead of minimizing this planning loss, while leaving the MCTS algorithm and neural architecture unchanged? To directly maximize the episode score, we use evolution strategies, a family of algorithms for zeroth-order blackbox optimization. We compare both approaches across multiple single-agent environments. Our experiments suggest that directly maximizing the episode score tends to outperform minimizing the planning loss.

## 1 Introduction

Lookahead search and reasoning is a central paradigm in artificial intelligence, and has a long history (Newell and Ernst, 1965; Hart et al., 1968; Nilsson, 1971; Hart et al., 1972; Lanctot et al., 2017; Brown et al., 2018). In many domains, planning at execution time significantly improves performance. In domains like Sokoban, Pacman, and 2048, all state-of-the-art approaches use some form of planning by the agent. Many planning approaches use *Monte Carlo Tree Search (MCTS)*, which iteratively grows a search tree from the current state, and does so asymmetrically according to the information seen so far. A prominent subfamily of approaches in this category are AlphaZero and its variants, which leverage function approximation via neural networks to learn good heuristic predictions of the values and action distributions at each state, which can be used to guide the tree search. AlphaZero (and its variants) train this prediction function by minimizing a *planning loss* consisting of the sum of a *value loss* and a *policy loss*.

In this paper, we set out to explore whether we can outperform AlphaZero and its variants in single-agent environments by *directly maximizing the episode score* instead, while leaving all other aspects of the agent, MCTS algorithm, and neural architecture unchanged. Since MCTS is not differentiable, to maximize the episode score, we employ evolution strategies, a family of algorithms for zeroth-order black-box optimization.

The structure of the paper is as follows. In §2, we present a detailed formulation of the problem. In §3, we describe related work. In §4, we present our method. In §5, we describe our experimental benchmarks and present our results. In §6, we discuss the experimental results. In §7, we present our conclusion and suggest directions for future research.

## 2 PROBLEM FORMULATION

In this section, we formulate the problem in detail and introduce notation. If $\mathcal{X}$ is a set, $\triangle\mathcal{X}$ denotes the set of probability distributions on $\mathcal{X}$. An *environment* is a tuple $(\mathcal{S}, \mathcal{A}, \rho, \delta)$ where $\mathcal{S}$ is a set of states, $\mathcal{A}$ is a set of actions, $\rho : \triangle\mathcal{S}$ is an initial state distribution, and $\delta : \mathcal{S} \times \mathcal{A} \to \mathbb{R} \times \mathbb{R} \times \mathcal{S}$ is a transition function. A *policy* is a function $\mathcal{S} \to \triangle\mathcal{A}$ that maps a state to an action distribution. Given an environment and policy, an *episode* is a tuple $(s, a, r, \gamma)$ that is generated as follows. First, an initial state $s_0 \sim \rho$ is sampled. Thereafter, on each timestep $t \in \mathbb{N}$, an action $a_t \sim \pi(s_t)$ is sampled, and a reward, discount factor, and new state $(r_t, \gamma_t, s_{t+1}) = \delta(s_t, a_t)$ are obtained. The discount factor represents the probability of the episode ending at that timestep. For a given episode, the *return* at timestep $t \in \mathbb{N}$ is defined recursively as $R_t = r_t + \gamma_t R_{t+1}$. The *score* is the return at the initial timestep, $R_0$. Our goal is to find a policy $\pi : \mathcal{S} \to \triangle\mathcal{A}$ that maximizes the expected score $\mathbb{E}\, R_0$.

## 3 RELATED WORK

In this section, we describe related work. Monte Carlo methods are a wide class of computational algorithms that use repeated random sampling to estimate numerical quantities. In the setting of planning, Monte-Carlo evaluation estimates the value of a position by averaging the return of several random rollouts. *Monte-Carlo Tree Search (MCTS)* (Coulom, 2007) combines Monte-Carlo evaluation with tree search. Instead of backing-up the min-max value close to the root, and the average value at some depth, it uses a more general backup operator that progressively changes from averaging to min-max as the number of simulations grows. MCTS grows the search tree asymmetrically, focusing on more promising subtrees.

AlphaGo (Silver et al., 2016) used a variant of MCTS to tackle the two-player board game of Go. It used a neural network to evaluate board positions *and* select moves. These networks are trained using a combination of supervised learning from human expert games and reinforcement learning from self-play. It was the first computer program to defeat a human professional player. AlphaGo Zero (Silver et al., 2017a) used reinforcement learning alone, *without* any human data, guidance or domain knowledge beyond game rules. AlphaZero (Silver et al., 2018) generalized AlphaGo Zero into a single algorithm that achieved superhuman performance in many challenging domains.

MuZero (Schrittwieser et al., 2020) combined AlphaZero's tree-based search with a *learned dynamics model*. The latter allows it to plan in environments where the agent does *not* have access to a simulator of the environment at execution time. Gumbel MuZero (Danihelka et al., 2022) is a policy improvement algorithm based on sampling actions without replacement. It replaces the more heuristic mechanisms by which AlphaZero selects actions at root and non-root nodes. Empirically, it yields significantly better performance when planning with few simulations.

MCTS is a state-of-the-art general-purpose technique for search, planning, and optimization in single-agent settings. For example, in the papers that introduced them, the prominent MCTS-based methods MuZero and Gumbel MuZero were shown to be state of the art in single-agent settings, including 57 different Atari games, the canonical video game environment for testing AI techniques. Świechowski et al. (2023) note that "Automated planning is one of the major domains of application of the MCTS algorithm outside games." Vallati et al. (2015) note that winning approaches of the International Probabilistic Planning Competition were using MCTS. This competition included combinatorial optimization problems, such as the minimization of open stacks problem (Yanasse and Senne, 2010).

MCTS has also been used in other discrete combinatorial problems, such as polynomial evaluation (Kuipers et al., 2013), low latency communication (Jia et al., 2020), generating large-scale floor plans with adjacency constraints (Shi et al., 2020), query selection in kidney exchange (McElfresh et al., 2020), and preference elicitation (Martin et al., 2024). Abe et al. (2019) used AlphaZero to solve NP-hard problems on graphs, including min vertex cover and max cut. Fawzi et al. (2022) used an AlphaZero-based algorithm, AlphaTensor, to discover efficient and provably-correct algorithms for multiplication of arbitrary matrices. Xu and Lieberherr (2019) showed that neural MCTS can be used in a general way to solve combinatorial optimization problems.

# 4 PROPOSED METHOD

In this section, we present a detailed description of our proposed method, which we call AlphaZeroES. The essential difference to AlphaZero is described in §4.3.

## 4.1 PLANNING ALGORITHM

We use the implementation of Gumbel MuZero (Danihelka et al., 2022), which is the prior state of the art for this setting, found in the open-source Google DeepMind library Mctx (DeepMind et al., 2020). It iteratively constructs a search tree starting from a state $s_0$. Each node in the tree contains a state, predicted value, predicted action probabilities, and, for each action, a visit count $N$, action value $Q$, reward, and discount factor. Each iteration of the algorithm consists of three phases: *selection*, *expansion*, and *backpropagation*.

During *selection*, we start at the root and traverse the tree until a leaf edge is reached. At internal nodes, we select actions according to the policy described in Danihelka et al. (2022). When we reach a leaf edge $(s, a)$, we perform *expansion* as follows. We compute $(r, \gamma, s') = \delta(s, a)$, storing $r$ and $\gamma$ in the edge's parent node. We then query the agent's *prediction function* $(v, p) = f_\theta(s')$ to obtain the predicted value and action probabilities of $s'$. A new node is added to the tree containing this information, with action visit counts and action values initialized to zero. Finally, we perform *backpropagation* as follows. The new node's value estimate is backpropagated up the tree to the root in the form of an $n$-step return. Specifically, from $t = T$ to $0$, where $T$ is the length of the trajectory, we compute an estimate of the cumulative discounted return $G_t$ that bootstraps from the value estimate $v$: $G_T = v$ and $G_t = r_t + \gamma_t G_{t+1}$. For each such $t$, we update the statistics for the edge corresponding to $(s_t, a_t)$ as follows: $Q(s_t, a_t) \leftarrow \frac{N(s_t,a_t)Q(s_t,a_t)+G_t}{N(s_t,a_t)+1}$, $N(s_t, a_t) \leftarrow N(s_t, a_t) + 1$. The *simulation budget* is the total number of iterations, which is the number of times the search tree is expanded, and therefore the size of the tree.

## 4.2 PREDICTION FUNCTION

The prediction function of the agent takes an environment state as input and outputs a probability distribution over actions and value estimate. Following Silver et al. (2018), we use a single neural network that outputs both of these. Our experimental settings have states that are naturally modeled as *sets* of objects (such as sets of cities, facilities, targets, boxes, etc.), where each object can be described by a vector (e.g., the coordinates of a city and whether it has been visited or not). Therefore, we seek a neural network architecture that can process a *set* of vectors, rather than just a single vector. Early works on neural networks for processing set inputs include McGregor (2007; 2008).

In our experiments, we use *DeepSets (Zaheer et al., 2017)*, a neural network architecture that can process sets of inputs in a way that is equivariant or invariant (depending on the desired type of output) with respect to the inputs. It is known to be a universal approximator for continuous set functions, provided that the model's latent space is sufficiently high-dimensional (Wagstaff et al., 2022). DeepSets may be viewed as the most efficient incarnation of the Janossy pooling paradigm (Murphy et al., 2018), and can be generalized by Transformers (Vaswani et al., 2017; Kim et al., 2021). A permutation-equivariant layer of the DeepSets architecture has the form (Zaheer et al., 2017, Supplement p. 19) $\mathbf{Y} = \sigma(\mathbf{X} \cdot \mathbf{A} + \mathbf{1} \otimes \mathbf{b} + \mathbf{1} \otimes ((\mathbf{1} \cdot \mathbf{X}) \cdot \mathbf{C}))$ where $\mathbf{X} \in \mathbb{R}^{n \times d}$, $\mathbf{Y} \in \mathbb{R}^{n \times k}$, $\mathbf{A}, \mathbf{C} \in \mathbb{R}^{n \times k}$, $\mathbf{b} \in \mathbb{R}^k$, and $\mathbf{1}$ is the all-ones vector of appropriate dimensionality, and $\sigma$ is a nonlinear activation function, such as ReLU. Here, $n$ is the size of the set (i.e., number of inputs/outputs), $d$ is the dimension of each input, and $k$ is the dimension of each output. A permutation-invariant layer is simply a permutation-equivariant layer followed by global average pooling (yielding an output that is a vector rather than a matrix) followed by a nonlinearity.

In problems where the action space matches the set of inputs (such as cities in the TSP problem, or points in the vertex $k$-center and maximum diversity problems), the predicted action logits are read out via a dense layer following the permutation-equivariant layer, before global pooling. In problems where the action space is a fixed set of actions (such as Sokoban and the navigation problems), the predicted action logits are read out via a dense layer following the permutation-invariant layer. In both cases, the predicted value is read out via a dense layer from the output of the permutation-invariant layer.

For clarity, we emphasize that we use *the exact same architecture* for both AlphaZero and AlphaZeroES in each problem. This is an apples-to-apples comparison. The only thing that changes is the optimization objective. AlphaZero itself is largely agnostic to the particular neural architecture available to the agent. It has been used in conjunction with simple feedforward networks, convolutional networks, attention-based networks (which encode permutation invariance), and so on.

### 4.3 TRAINING PROCEDURE

We are now ready to present the essential difference between AlphaZero and our AlphaZeroES. The difference lies in the training objective, which in turn entails a difference in the training procedure. AlphaZero minimizes a *planning loss*, which is the sum of a value loss $\sum_t (R_t - v_t)^2$ and a *policy loss* $\sum_t \mathrm{H}(w_t, p_t)$. Here, $(v_t, p_t) = f_\theta(s_t)$ is the predicted state value and action probabilities for $s_t$, respectively. $(R_t - v_t)^2$ is the squared difference between $v_t$ and the actual episode return $R_t$. $\mathrm{H}(w_t, p_t)$ is the cross entropy between the action weights $w_t$ returned by the MCTS algorithm for $s_t$ and $p_t$. Our approach keeps *exactly the same* architecture, hyperparameters, and MCTS algorithm as AlphaZero, but changes the optimization objective. Specifically, instead of minimizing the planning loss, we *directly maximize the episode score*. The parameters that are optimized are exactly those of AlphaZero, namely, the neural network parameters of the prediction function. Only the training objective is different.

One way to directly optimize the episode score is to use policy gradient methods, which yield an estimator of the gradient of the expected return with respect to the agent's parameters. There is a vast literature on policy gradient methods, which include REINFORCE (Williams, 1992) and actor-critic methods (Konda and Tsitsiklis, 1999; Grondman et al., 2012). However, there is a problem. Most of these methods assume that the policy is *differentiable*—more precisely, that its output action distribution is differentiable with respect to the parameters of the policy. However, our planning policy uses MCTS as a subroutine, and standard MCTS is not differentiable. Because our policy contains a non-differentiable submodule, we need to find an alternative way to optimize the policy's parameters. Furthermore, Metz et al. (2021) show that differentiation can fail to be useful when trying to optimize certain functions—specifically, when working with an iterative differentiable system with chaotic dynamics. Fortunately, we can turn to black-box (i.e., zeroth-order) optimization. Black-box optimization uses only function evaluations to optimize a function with respect to a set of inputs. In particular, it does not require gradients. In our case, the black-box function maps our policy's parameters to a sampled episode score.

There is a class of black-box optimization algorithms called *evolution strategies (ES)* (Rechenberg and Eigen, 1973; Schwefel, 1977; Rechenberg, 1978) that maintain and evolve a population of parameter vectors. *Natural evolution strategies (NES)* (Wierstra et al., 2014; Yi et al., 2009) represent the population as a distribution over parameters and maximize its average objective value using the score function estimator. For many parameter distributions, such as Gaussian smoothing, this is equivalent to evaluating the function at randomly-sampled points and estimating the gradient as a sum of estimates of directional derivatives along random directions (Duchi et al., 2015; Nesterov and Spokoiny, 2017; Shamir, 2017; Berahas et al., 2022). ES can be used to learn non-differentiable parameters of large supervised models, such as sparsity masks for weights (Lenc et al., 2019).

We use OpenAI-ES (Salimans et al., 2017), an NES algorithm that has been shown to be effective for reinforcement learning (Salimans et al., 2017), including training large language models (Qiu et al., 2025). It is based on the identity $\nabla_{\mathbf{x}} \mathrm{E}_{\mathbf{z} \sim \mathcal{D}} f(\mathbf{x} + \sigma \mathbf{z}) = \frac{1}{\sigma} \mathrm{E}_{\mathbf{z} \sim \mathcal{D}} f(\mathbf{x} + \sigma \mathbf{z}) \mathbf{z}$, where $\mathcal{D}$ is the standard multivariate normal distribution. This algorithm is shown in Algorithm 1. Like Salimans et al. (2017), we use antithetic sampling (Geweke, 1988), also called mirrored sampling (Brockhoff et al., 2010), to reduce variance. It samples antithetic pairs of perturbations $(\mathbf{z}_i, -\mathbf{z}_i)$.

This algorithm is massively parallelizable, since each $\delta_i$ can be evaluated on a separate worker. Furthermore, communication between workers is minimal. All workers are initialized with the same random seed. Worker $i$ evaluates $\delta_i$, sends it to the remaining workers, and receives the other workers' values (this is called an all-gather operation in distributed computing). Thus the workers compute the same $\mathbf{g}$ and stay synchronized. Again, each worker computes the $\delta_i$ corresponding to *its own* index $i$ and receives the others from the other workers, but generates the all workers' perturbation *vectors* $\{\mathbf{z}_j\}_{j \in \mathcal{I}}$ itself, which is more efficient than communicating them. The shared random seed

---

**Algorithm 1** Evolution strategies (with a vanilla SGD optimizer).

---

**Input:** Initial parameters $\mathbf{x} \in \mathbb{R}^d$, noise scale $\sigma \in \mathbb{R}$, learning rate $\alpha \in \mathbb{R}$, set of workers $\mathcal{I}$.
**for** $t = 0, 1, 2, \ldots$ **do**
    Sample perturbations $\mathbf{z}_1, \ldots, \mathbf{z}_n \sim \mathcal{N}(\mathbf{0}_d, I_d)$
    For each $i \in \mathcal{I}$, let worker $i$ compute $\delta_i \leftarrow f(\mathbf{x} + \sigma \mathbf{z}_i)$
    Compute pseudogradient $\mathbf{g} \leftarrow \frac{1}{\sigma |\mathcal{I}|} \sum_{i \in \mathcal{I}} \delta_i \mathbf{z}_i$
    Update parameters $\mathbf{x} \leftarrow \mathbf{x} + \alpha \mathbf{g}$

---

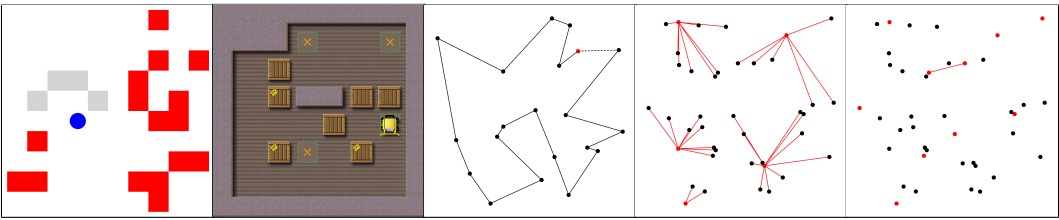

Figure 1: Example states for each environment: Navigation, Sokoban, TSP, VKCP, and MDP.

ensures that workers can compute identical perturbation vectors without communication. The only worker-dependent computation is $\delta_i$.

Notably, AlphaZeroES needs only the parameter perturbation vector $\mathbf{z}$ and the final episode score to update the parameters. In contrast, AlphaZero needs to compute gradients of the parameters via backpropagation (reverse-mode automatic differentiation) through the neural network and over the timesteps of the episode. In our experiments, AlphaZero and AlphaZeroES took about the same amount of time per iteration.

## 5 EXPERIMENTS

In this section, we describe our experiments. We use 10 trials per experiment, 1000 episodes per batch (for both training and evaluation at the end of each epoch), 1000 training batches per epoch, 4 hours of training time per trial, the AdaBelief (Zhuang et al., 2020) optimizer[1], a perturbation scale of 0.1 for OpenAI-ES, an MCTS simulation budget of 8,[2] hidden layer sizes of 16 for the DeepSets network, 1 equivariant plus 1 invariant hidden layer for the DeepSets network, and the ReLU activation function. We used an NVIDIA A100 SXM4 40GB GPU. Each trial uses 1 such GPU all to itself. This keeps the comparison between AlphaZero and AlphaZeroES as precise as possible. For our code, we use Python 3.12.2, JAX 0.4.28 (Bradbury et al., 2018), Flax 0.8.3 (Heek et al., 2024), Optax 0.2.2 (DeepMind et al., 2020), Mctx 0.0.5 (DeepMind et al., 2020), and Matplotlib 3.8.4 (Hunter, 2007). In our plots, we show the episode scores attained by AlphaZero (labeled es=0 in the plot legend) vs. AlphaZeroES (labeled es=1 in the plot legend). At any point along the X axis, AlphaZero and AlphaZeroES have undergone the same number of episodes of learning. To perform a fair comparison, since AlphaZero and AlphaZeroES optimize different objectives, we test both across a wide range of learning rates (labeled lr in the plot legend). In addition, we show value and policy losses over the course of training. Though AlphaZeroES does not optimize these losses directly, we wish to observe what happens to them as a side-effect of maximizing the episode score. Solid lines show the mean across trials, and bands show the standard error of the mean. Our goal is not to develop the best special-purpose solver for any one of these domains. Rather, we are interested in a *general*-purpose approach that can tackle *all* of these domains and learn good heuristics on its own. Due to space constraints, we relegate the plots showing value and policy loss to the appendix.

---

[1]Both AlphaZero and AlphaZeroES can be combined with any optimizer from the literature. Finding the best optimizer is not the focus of this paper. AdaBelief is a well-known optimizer with many citations. We chose it because it is (a) relatively well-known and (b) outperforms SGD and Adam.

[2]Gumbel Muzero, the AlphaZero variant we use, can learn reliably with as few as 2 simulations, and was evaluated in its paper with 2, 4, and 16 simulations (Danihelka et al., 2022, p. 8).

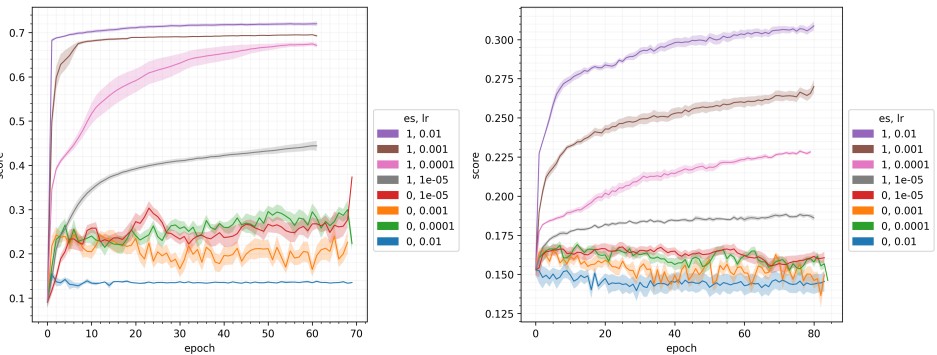

Figure 2: Navigation score.          Figure 3: Sokoban score.

## 5.1 NAVIGATION

In this environment, an agent navigates a gridworld to reach as many targets as possible within a given time limit. At the beginning of each episode, targets are placed uniformly at random in a $10 \times 10$ grid, as is the agent. On each timestep, the agent can move up, down, left, or right by one tile. The agent reaches a target when it moves into the same tile. The agent receives a reward of $+0.05$ when it reaches a target. Thus the agent is incentivized to reach as many targets as possible within the time limit. For our experiments, we use 20 targets and a time limit of 50 steps. The prediction network observes a set of vectors, one for each target, where each vector contains the coordinates of the target, a boolean 0-1 flag indicating whether it has already been reached, and the number of episode timesteps remaining. This environment has been used before as a benchmark by Oh et al. (2017, §4.2). It resembles a traveling salesman-like problem in which several "micro" actions are required to perform the "macro" actions of moving from one city to another. (Also, the agent can visit cities multiple times and does not need to return to its starting city.) This models situations where several fine-grained actions are required to perform relevant tasks, such as moving a unit in a real-time strategy game a large distance across the map.

An example state is shown in Figure 1. The blue circle is the agent. Red squares are unreached targets. Gray squares are reached targets. Experimental results are shown in Figure 2 and 7. AlphaZeroES dramatically outperforms AlphaZero. Unlike AlphaZero, it does not seem to minimize the value and policy losses by a noticeable amount. In fact, for AlphaZeroES, the value and policy losses seem to *increase* over time as training proceeds (and the mean episode score increases). This will be a recurring pattern across environments, as we will observe with the other benchmarks. This phenomenon suggests that maximizing "self-consistency" via planning loss minimization, as standard AlphaZero does, is not necessarily aligned as an objective with performing better in the environment, as measured by mean episode score.

## 5.2 SOKOBAN

Sokoban is a puzzle in which an agent pushes boxes around a warehouse to get them to storage locations. It is played on a grid of tiles. Each tile may be a floor or a wall, and may contain a box or the agent. Some floor tiles are marked as storage locations. The agent can move horizontally or vertically onto empty tiles. The agent can also move a box by walking up to it and push it to the tile beyond, if the latter is empty. Boxes cannot be pulled, and they cannot be pushed to squares with walls or other boxes. The number of boxes equals the number of storage locations. The puzzle is solved when all boxes are placed at storage locations. Planning ahead is crucial, since an agent can easily get stuck if it makes the wrong move. Sokoban has been studied in the field of computational complexity and shown to be PSPACE-complete (Culberson, 1997). It has received significant interest in artificial intelligence research because of its relevance to automated planning (e.g., for autonomous robots), and is used as a benchmark. Sokoban's large branching factor and search tree depth contribute to its difficulty. Skilled human players rely mostly on heuristics and can quickly discard several futile or redundant lines of play by recognizing patterns and subgoals, narrowing down the search significantly. Various automatic solvers have been developed in the literature (Junghanns and Schaeffer, 1997;

2001; Froleyks and Balyo, 2016; Shoham and Schaeffer, 2020), many of which rely on heuristics, but more complex Sokoban levels remain a challenge.

Our environment is as follows. We use the unfiltered Boxoban training set (Guez et al., 2019), which contains 900,000 levels of size $10 \times 10$ each. At the beginning of each episode, we sample a level from this dataset. As a form of data augmentation, we sample one of the eight symmetries of the square (a horizontal flip, vertical flip, and/or 90-degree rotation) and apply it to the level. In each timestep, the agent has four actions available to it, for motion in each of the four cardinal directions. The level ends after a specified number of timesteps. (We use 50 timesteps.) The return at the end of an episode is the number of goals that are covered with boxes. Thus the agent is incentivized to cover all of the goals. The prediction network observes a set of vectors, one for each tile in the level, where each vector contains the 2 coordinates of the tile, 4 boolean flags indicating whether the tile contains a wall, goal, box, and/or agent, and the number of episode timesteps remaining. An example state is shown in Figure 1. This was rendered by JSoko (Meger, 2023), an open-source Sokoban implementation. The yellow vehicle is the agent, who must push the brown boxes into the goal squares marked with Xs. (Boxes tagged "OK" are on top of goal squares.) Experimental results are shown in Figure 3 and 8. AlphaZeroES dramatically outperforms AlphaZero. Unlike AlphaZero, it does not seem to minimize the value and policy losses by a noticeable amount.

### 5.3 TSP

The *traveling salesman problem (TSP)* is a classic combinatorial optimization problem. Given a set of cities and their pairwise distances, the goal is to find a shortest route that visits each city once and returns to the starting city. This problem has important applications in operations research, including logistics, computer wiring, vehicle routing, and various other planning problems (Matai et al., 2010). TSP is known to be NP-hard (Karp, 1972), even in the Euclidean setting (Papadimitriou, 1977). Various approximation algorithms and heuristics (Nilsson, 2003) have been developed for it. Our environment is as follows. We seek to learn to solve TSP in general, not just one particular instance of it. Thus, on every episode, a new problem instance is generated by sampling a matrix $\mathbf{X} \sim \mathrm{Uniform}([0, 1]^{n \times 2})$, representing a sequence of $n \in \mathbb{N}$ cities. In our experiments, we use $n = 20$. At timestep $t \in [n]$, the agent chooses a city $a_t \in [n]$ that has not been visited yet. At the end of the episode, the length of the tour through this sequence of cities (including the segment from the final city to the initial one) is computed, and treated as the *negative* score. Thus the agent is incentivized to find the shortest tour through all the cities. Formally, the final score is $-\sum_{t \leq n} d(\mathbf{X}_{a_t}, \mathbf{X}_{a_{t+1} \bmod n})$, where $d$ is the Euclidean metric. The prediction network observes a set of vectors, one for each city, where each vector contains the coordinates of the city and 3 boolean 0-1 flags indicating whether it has already been visited, whether it is the initial city, and whether it is the current city.

An example state is shown in Figure 1. Dots are cities. The red dot is the initial city. The lines connecting the dots constitute the constructed path. The dotted line is the last leg from the final city back to the initial city. Experimental results are shown in Figure 4 and 9. AlphaZeroES dramatically outperforms AlphaZero. Interestingly, as a side effect, it minimizes the policy loss about as much as AlphaZero does. It also minimizes the value loss (except at the highest learning rate), though to a lesser extent than AlphaZero.

### 5.4 VKCP

The *vertex k-center problem (VKCP)* is a classic combinatorial optimization problem that has applications in facility location and clustering. The problem is as follows. Given $n$ points in $\mathbb{R}^d$, select a subset $\mathcal{S}$ of $k$ points that minimizes the distance from any point in the original set to its nearest point in $\mathcal{S}$. The $n$ points can be interpreted as possible locations in which to build facilities (e.g., fire stations, police stations, supply depots, etc.), where $\mathcal{S}$ is the set of locations in which such facilities are built, and the goal is to minimize the maximum distance from any location to its nearest facility. (There is also a variant of the problem that seeks to minimize the *mean* distance.) This problem was first proposed by Hakimi (1964). It is NP-hard, and various approximation algorithms have been proposed for it (Kariv and Hakimi, 1979; Gonzalez, 1985; Dyer and Frieze, 1985; Hochbaum and Shmoys, 1985; Shmoys, 1994). A survey and evaluation of approximation algorithms can be found in Garcia-Diaz et al. (2019).

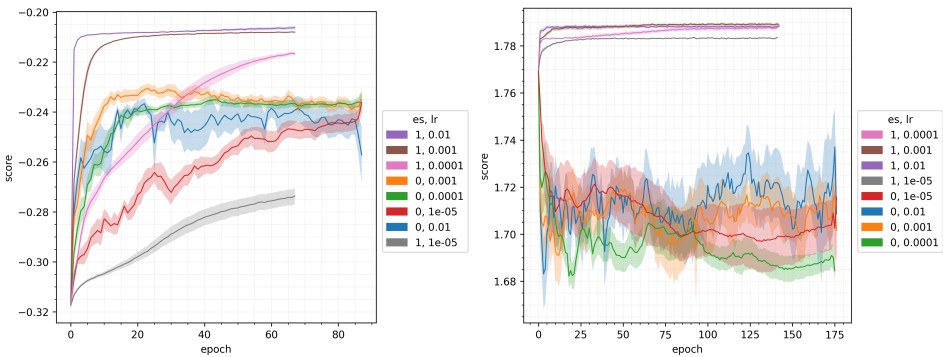

Figure 4: TSP score.    Figure 5: VKCP score.

We sample $n = 40$ locations uniformly at random from the unit square and let $k = 20$. At any timestep $t$, the agent selects a location $a_t \in [n]$ that has not been selected yet to add a facility at that location. The final score is $-\max_{i \in [n]} \min_{j \in \mathcal{S}} d(\mathbf{x}_i, \mathbf{x}_j)$, where $\mathbf{x}_i \in [0, 1]^2$ is the position of point $i \in [n]$ and $d$ is the Euclidean metric. The prediction network observes a set of vectors, one for each point, where each vector contains the coordinates of the point and a single bit indicating whether it is in the subset $\mathcal{S}$. An example state is shown in Figure 1. Black dots are locations, red dots are facilities placed so far, and red lines connect locations to their nearest facility. Experimental results are shown in Figure 5 and 10. AlphaZeroES dramatically outperforms AlphaZero. In this environment, AlphaZeroES hardly minimizes the value and policy losses as a side effect.

## 5.5 MDP

In the *maximum diversity problem (MDP)*, we are given $n$ points in $\mathbb{R}^d$, and we are asked to select a subset $\mathcal{S}$ of $k$ points that maximizes the minimum distance between distinct points. (There is also a variant of the problem that seeks to maximize the *mean* distance between distinct points.) This problem is strongly NP-hard, as can be shown via reduction from the clique problem (Kuo et al., 1993; Ghosh, 1996). Various heuristics have been proposed for it (Glover et al., 1998; Katayama and Narihisa, 2005; Silva et al., 2007; Duarte and Martí, 2007; Martí et al., 2010; Lozano et al., 2011; Wu and Hao, 2013; Martí et al., 2013). This problem has applications in ecology, medical treatment, genetic engineering, capital investment, pollution control, system reliability, telecommunication services, molecular structure design, transportation sys-

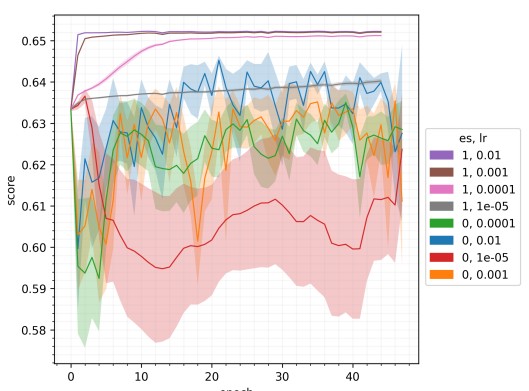

Figure 6: MDP score.

tem control, emergency service centers, and energy options, as cataloged by Glover et al. (1998, Table 1).

For our experiments, we sample $n = 40$ locations uniformly at random from the unit square and let $k = 20$. At any timestep $t$, the agent can select a point $a_t \in [n]$ that has not been selected yet to add to the set $\mathcal{S}$. The final score is $\min_{i, j \in \mathcal{S}, i \neq j} d(\mathbf{x}_i, \mathbf{x}_j)$, where $\mathbf{x}_i \in [0, 1]^2$ is the position of point $i$ and $d$ is the Euclidean metric. The prediction network observes a set of vectors, one for each point, where each vector contains the coordinates of the point and a bit flag indicating whether it has been included in the set. An example state is shown in Figure 1. Black dots are points, red dots are points selected so far, and the red line connects the closest pair of points in the set selected so far. Experimental results are shown in Figure 6 and 11. AlphaZeroES dramatically outperforms AlphaZero. As a side effect, it minimizes the policy loss about as much as AlphaZero does. However, unlike AlphaZero, it does not seem to minimize the value loss.

## 6 DISCUSSION

**Why does our method work?**   Our method did not drive value and policy losses down to zero, as standard AlphaZero does, suggesting that maximizing "self-consistency" is not necessarily required to perform better in the environment in terms of score. One reason might be that optimal or strong performance does not actually require *internal consistency* (of value and action predictions), and achieving *good performance* might be easier than achieving internal consistency.

There are situations where learning a good policy is easy, but learning a good function is hard. Consider an environment where there is a simple optimal policy, but the value function under that policy is complicated—that is, for any given state, it is easy to determine what the "right" action to take is, but difficult to predict the final return. AlphaZero's performance intrinsically depends on the accuracy of its learned value function, since that value function is used as an oracle inside the MCTS algorithm in a way that ultimately determines what action to take. If this value function is difficult to learn, AlphaZero might struggle. In fact, even being *semi*-accurate with respect to values does not, in and of itself, guarantee good action selection. The value estimates also need to be *order*-accurate—that is, accurate with respect to their relative rankings or differences—since this ultimately determines which actions MCTS chooses.

On the other hand, AlphaZeroES has the flexibility to simply optimize a policy directly, even if it has not learned an accurate value function for it. The value function being accurate might be helpful, but is not necessary. In summary, direct policy methods sometimes succeed where value-based methods fail. This can happen when a good policy is more easily representable (and learnable) than a good value function. In those cases, direct policy improvement can easily yield a good policy. Conversely, relying on a poorly-approximated critic can actually *hamper* performance. To illustrate this point, in the appendix, we give concrete examples of *simple* environments where AlphaZero fails while AlphaZeroES succeeds.   In the appendix, we also include an ablation study that investigates whether the improvement of AlphaZeroES over AlphaZero comes mostly from an improved value output or an improved policy output. Interestingly, the answer is environment-dependent.

## 7 CONCLUSIONS AND FUTURE RESEARCH

In this paper, we set out to study whether AlphaZero and its newest variants can be improved by maximizing the episode score directly instead of minimizing the standard planning loss. Since MCTS is not differentiable, we maximize the episode score by using evolution strategies. We conducted experiments across multiple domains, including standard combinatorial optimization problems and motion planning problems from the literature. In each setting, our approach yielded a dramatic improvement in performance over planning loss minimization.

Our work opens up new possibilities for tackling environments where planning is important. It does this by allowing agents to learn to leverage internal nondifferentiable planning algorithms, such as MCTS, *in a purely blackbox way* that does not depend on the internal details of those algorithms. Instead of training the agent's parameters to minimize some indirect proxy objective, such as a planning loss, we can now maximize the desired objective *directly*.

**Limitations**   The original AlphaZero and Gumbel MuZero MCTS algorithms are designed for fully-observable deterministic environments. Thus, so is our method. An extension to stochastic environments exists in the form of Stochastic MuZero (Antonoglou et al., 2022). By replacing the MCTS algorithm with that of Stochastic MuZero, it might be possible to extend our method to stochastic environments. Another potential direction for future research might be to extend our work to adversarial or multiagent settings. Doing so would require introducing concepts from game theory and making modifications to our method. For example, our method uses ES to maximize the episode score. However, solving a two-player zero-sum game is not a pure *maximization* problem, but rather a *min-max* (saddle-point) problem. Solving such a problem requires more sophisticated gradient dynamics. It might be possible to use a modified version of ES to seek equilibria of the players' individual episode scores with respect to their parameters. Related works for this include Bichler et al. (2021), Martin and Sandholm (2023), and Martin and Sandholm (2025). This is outside the scope of this paper, but potentially interesting for future research.

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

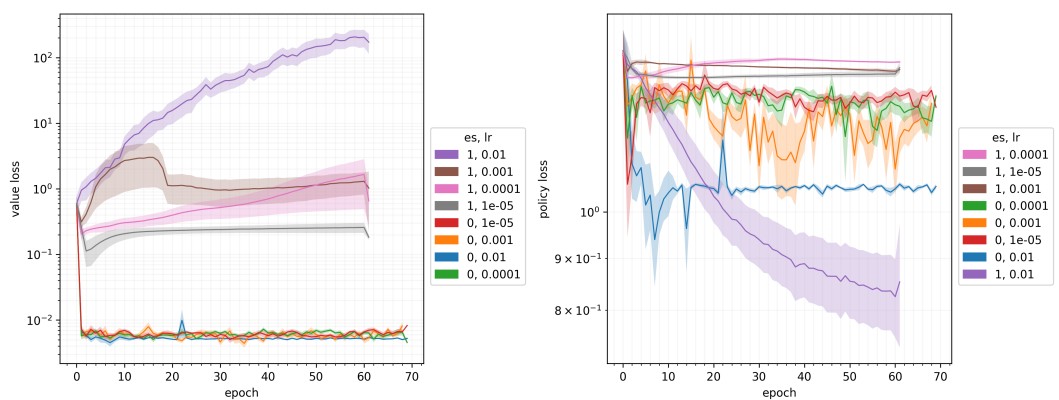

Figure 7: Navigation losses.

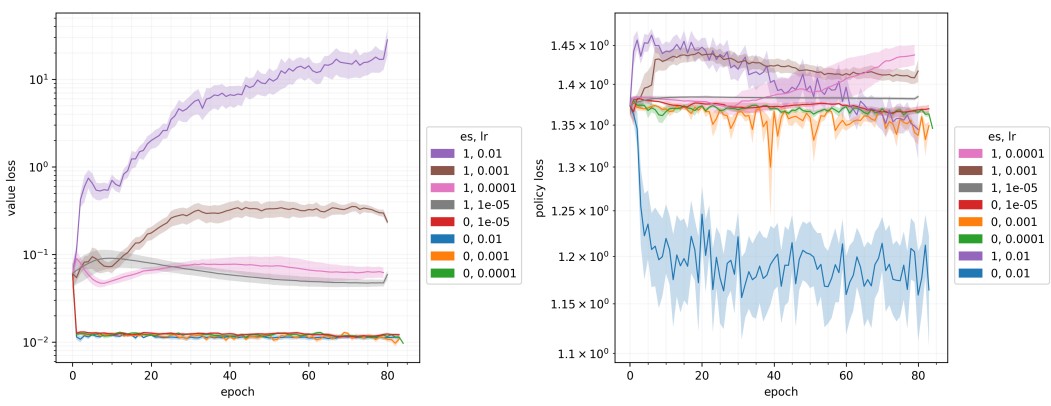

Figure 8: Sokoban losses.

## A ADDITIONAL FIGURES

In this section, we include additional figures that did not fit in the body of the paper.

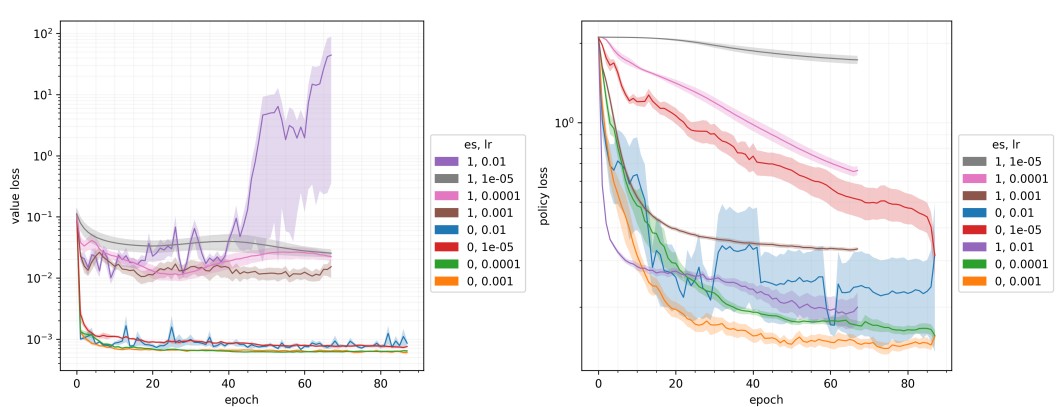

Figure 9: TSP losses.

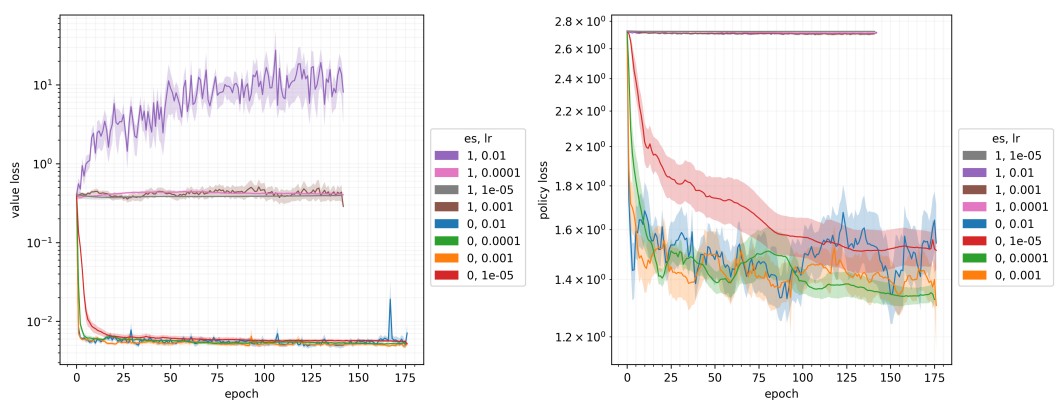

Figure 10: VKCP losses.

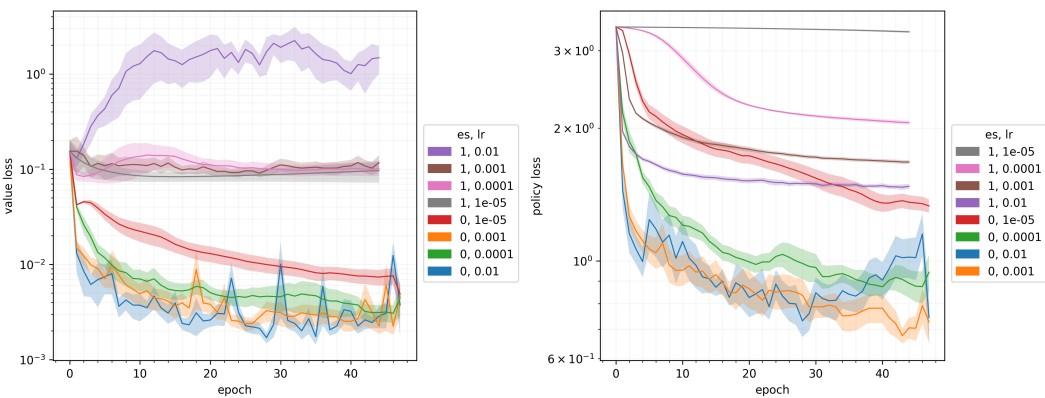

Figure 11: MDP losses.

## B ADDITIONAL RELATED WORK

In this section, we include additional related work that did not fit in the body of the paper.

### B.1 AGENTS THAT USE NEURAL NETWORKS AND PLANNING

*Value Iteration Network (VIN)* (Tamar et al., 2016) is a fully differentiable network with a planning module embedded within. It can learn to plan and predict outcomes that involve planning-based reasoning, such as policies for reinforcement learning. It uses a differentiable approximation of the value-iteration algorithm, which can be represented as a convolutional network, and is trained end-to-end using standard backpropagation.

Predictron (Silver et al., 2017b) consists of a fully abstract model, represented by a Markov reward process, that can be rolled forward multiple "imagined" planning steps. Each forward pass accumulates internal rewards and values over multiple planning depths. The model is trained end-to-end so as to make these accumulated values accurately approximate the true value function.

*Value Prediction Network (VPN)* (Oh et al., 2017) integrates model-free and model-based RL methods into a single network. In contrast to previous model-based methods, it learns a dynamics model with abstract states that is trained to make action-conditional predictions of future returns rather than future observations. VIN performs value iteration over the entire state space, which requires that 1) the state space is small and representable as a vector with each dimension corresponding to a separate state and 2) the states have a topology with local transition dynamics (such as a 2D grid). VPN does not have these limitations. VPN is trained to make its predicted values, rewards, and discounts match up with those of the real environment (Oh et al., 2017, §3.3).

*Imagination-Augmented Agent (I2A)* (Racanière et al., 2017) augments a model-free agent with imagination by using environment models to simulate imagined trajectories, which are provided as additional context to a policy network. An environment model is any recurrent architecture which can be trained in an unsupervised fashion from agent trajectories. Given a past state and current action, the environment model predicts the next state and observation. The imagined trajectory is initialized with the current observation and rolled out multiple time steps into the future by feeding simulated observations.

MCTSnet (Guez et al., 2018) incorporates simulation-based search inside a neural network, by expanding, evaluating and backing-up a vector embedding. The parameters of the network are trained end-to-end using gradient-based optimization. When applied to small searches in the well-known planning problem Sokoban, it outperformed prior MCTS baselines.

TreeQN (Farquhar et al., 2018) is an end-to-end differentiable architecture that substitutes value function networks in discrete-action domains. Instead of directly estimating the state-action value from the current encoded state, as in *Deep Q-Networks (DQN)* (Mnih et al., 2015), it uses a learned dynamics model to perform planning up to some fixed-depth. The result is a recursive, tree-structured network between the encoded state and the predicted state-action values at the leafs. The authors also propose ATreeC, an actor-critic variant that augments TreeQN with a softmax layer to form a stochastic policy network. Unlike MCTS-based methods, the shape of the planning tree is fixed, and the agent cannot "focus" on more promising subtrees to expand during planning.

Yang et al. (2020) proposed Continuous MuZero, an extension of MuZero to continuous actions, and showed that it outperforms the *soft actor-critic (SAC)* algorithm. Hubert et al. (2021) proposed Sampled MuZero, an extension of the MuZero algorithm that is able to learn in domains with arbitrarily complex action spaces (including ones that are continuous and high-dimensional) by planning over sampled actions.

Stochastic MuZero (Antonoglou et al., 2022) extended MuZero to environments that are inherently stochastic, partially observed, or so large and complex that they appear stochastic to a finite agent. It learns a stochastic model incorporating after-states following an action, and uses this model to perform a stochastic tree search. It matches or exceeds the state of the art in a canonical set of environments, including 2048.

### B.2 MACHINE LEARNING FOR TUNING INTEGER PROGRAMMING AND COMBINATORIAL OPTIMIZATION SOLVERS

Another, different, form of learning in search techniques is tuning *integer programming (IP)* and *combinatorial optimization (CO)* (Schrijver, 2003) techniques. The idea of automated algorithm tuning goes back at least to Rice (1976). It has been applied in industrial practice at least since 2001, when Sandholm (2013) started using machine learning to learn IP algorithm configurations (related to branching, cutting plane generation, *etc.*) and IP formulations based on problem instance features, in the context of combinatorial auction winner determination in large-scale sourcing auctions. In 2007, the leading commercial general-purpose IP solvers started shipping with such automated configuration tools.

IP solvers typically use a tree search algorithm called branch-and-cut. However, such solvers typically come with a variety of tunable parameters that are challenging to tune by hand. Research has demonstrated the power of using a data-driven approach to automatically optimize these parameters.

Similarly, real-world applications that can be formulated as CO problems often have recurring patterns or structure that can be exploited by heuristics. The design of good heuristics or approximation algorithms for NP-hard CO problems often requires significant specialized knowledge and trial-and-error, which can be a challenging and tedious process.

The rest of this section reviews some of the newer work on automated algorithm configuration in IP and CO.

Khalil et al. (2017) sought to automate the CO tuning process using a combination of reinforcement learning and graph embedding. They applied their framework to a diverse range of optimization problems over graphs, learning effective algorithms for the Minimum Vertex Cover, Maximum Cut and Traveling Salesman problems.

Bengio et al. (2021) surveyed recent attempts from the machine learning and operations research communities to leverage machine learning to solve IP and CO problems. According to the authors, "Given the hard nature of these problems, state-of-the-art algorithms rely on handcrafted heuristics for making decisions that are otherwise too expensive to compute or mathematically not well defined. Thus, machine learning looks like a natural candidate to make such decisions in a more principled and optimized way." They cite Larsen et al. (2018), who train a neural network to predict the solution of a stochastic load planning problem for which a deterministic mixed integer linear programming formulation exists. The authors state that "The nature of the application requires to output solutions in real time, which is not possible either for the stochastic version of the load planning problem or its deterministic variant when using state-of-the-art MILP solvers. Then, ML turns out to be suitable for obtaining accurate solutions with short computing times because some of the complexity is addressed offline, *i.e.*, in the learning phase, and the run-time (inference) phase is extremely quick."

Another survey of reinforcement learning for CO can be found in Mazyavkina et al. (2021). According to the authors, "Many traditional algorithms for solving combinatorial optimization problems involve using hand-crafted heuristics that sequentially construct a solution. Such heuristics are designed by domain experts and may often be suboptimal due to the hard nature of the problems. *Reinforcement learning (RL)* proposes a good alternative to automate the search of these heuristics by training an agent in a supervised or self-supervised manner."

To address the scalability challenge in large-scale CO, Qiu et al. (2022) propose an approach called *Differentiable Meta Solver (DIMES)*. Unlike previous deep reinforcement learning methods, which suffer from costly autoregressive decoding or iterative refinements of discrete solutions, DIMES introduces a compact continuous space for parameterizing the underlying distribution of candidate solutions. Such a continuous space allows stable REINFORCE-based training and fine-tuning via massively parallel sampling.

Aironi et al. (2024) proposed a graph-based neural approach to linear sum assignment problems, which are well-known CO problems with applications in domains such as logistics, robotics, and telecommunications. In general, obtaining an optimal solution to such problems is computationally infeasible even in small settings, so heuristic algorithms are often used to find near-optimal solutions. Their paper investigated a general-purpose learning strategy that uses a bipartite graph to describe the problem structure and a message-passing graph neural network model to learn the correct mapping.

The proposed graph-based solver, although sub-optimal, exhibited the highest scalability, compared with other state-of-the-art heuristic approaches.

Georgiev et al. (2024) note that "Solving NP-hard/complete combinatorial problems with neural networks is a challenging research area that aims to surpass classical approximate algorithms. The long-term objective is to outperform hand-designed heuristics for NP-hard/complete problems by learning to generate superior solutions solely from training data." The authors proposed leveraging recent advancements in neural algorithmic reasoning to improve learning of CO problems.

Balcan et al. (2024) provide the first sample complexity guarantees for tree search parameter tuning, bounding the number of samples sufficient to ensure that the average performance of tree search over the samples nearly matches its future expected performance on the unknown instance distribution. Balcan et al. (2021) prove the first guarantees for learning high-performing cut-selection policies tailored to the instance distribution at hand using samples. Balcan et al. (2022) derive sample complexity guarantees for using machine learning to determine which cutting planes to apply during branch-and-cut.

## C  STATISTICAL TESTS

We show statistical tests for each environment in Table 1. For each environment's comparison, we selected the best-performing learning rate for each method (AlphaZero vs. AlphaZeroES) under 10 trials, and compare the final mean scores. We used the same JAX PRNG key for each individual pair, that is, common random numbers.

| Environment | Wilcoxon signed-rank test | | Paired t-test | |
| --- | --- | --- | --- | --- |
| | statistic | p-value | statistic | p-value |
| Navigation | 55 | 0.000976562 | 24.1637 | $8.51516 \times 10^{-10}$ |
| Sokoban | 55 | 0.000976562 | 24.3562 | $7.93596 \times 10^{-10}$ |
| TSP | 55 | 0.000976562 | 6.89033 | $3.57182 \times 10^{-5}$ |
| VKCP | 55 | 0.000976562 | 13.4227 | $1.47451 \times 10^{-7}$ |
| MDP | 55 | 0.000976562 | 3.85802 | 0.00192935 |

Table 1: Statistical tests for each environment.

All pairwise differences were positive, so the Wilcoxon statistic maxed out at $n(n+1)/2 = 10 \times 11/2 = 55$. All p-values are well under 0.05. In conclusion, all the results are highly statistically significant.

## D  SCALABILITY

In this section, we run experiments that test the scalability of our method, AlphaZeroES, in comparison to standard AlphaZero. Specifically, we see which method performs best for various problem sizes (such as number of nodes for TSP problems). Each individual run received exactly 1 hour of training time on a single NVIDIA A100 SXM4 40GB GPU. Results are shown in Figures 12, 13, and 14. In the legends of these plots, `loss=alphazero` denotes AlphaZero and `loss=score_es` denotes AlphaZeroES. Likewise, in Figure 15, we compare the scalability of AlphaZero against AlphaZeroES in terms of the size of the network (specifically, the hidden layer size). In all figures, AlphaZeroES outperforms AlphaZero regardless of the scale of the problem.

Regarding the performance of OpenAI-ES vs. classical gradient-based methods on high-dimensional problems, Salimans et al. (2017) note the following: "The resemblance of ES to finite differences suggests the method will scale poorly with the dimension of the parameters $\theta$. [...] However, it is important to note that this does not mean that larger neural networks will perform worse than smaller networks when optimized using ES: **what matters is the difficulty, or intrinsic dimension, of the optimization problem** [emphasis added]. To see that the dimensionality of our model can be completely separate from the effective dimension of the optimization problem, consider a regression problem where we approximate a univariate variable $y$ with a linear model $\hat{y} = \mathbf{x} \cdot \mathbf{w}$: if we double the number of features and parameters in this model by concatenating x with itself (i.e. using features

$\mathbf{x}' = (\mathbf{x}, \mathbf{x})$), the problem does not become more difficult. The ES algorithm will do exactly the same thing when applied to this higher dimensional problem, as long as we divide the standard deviation of the noise by two, as well as the learning rate."

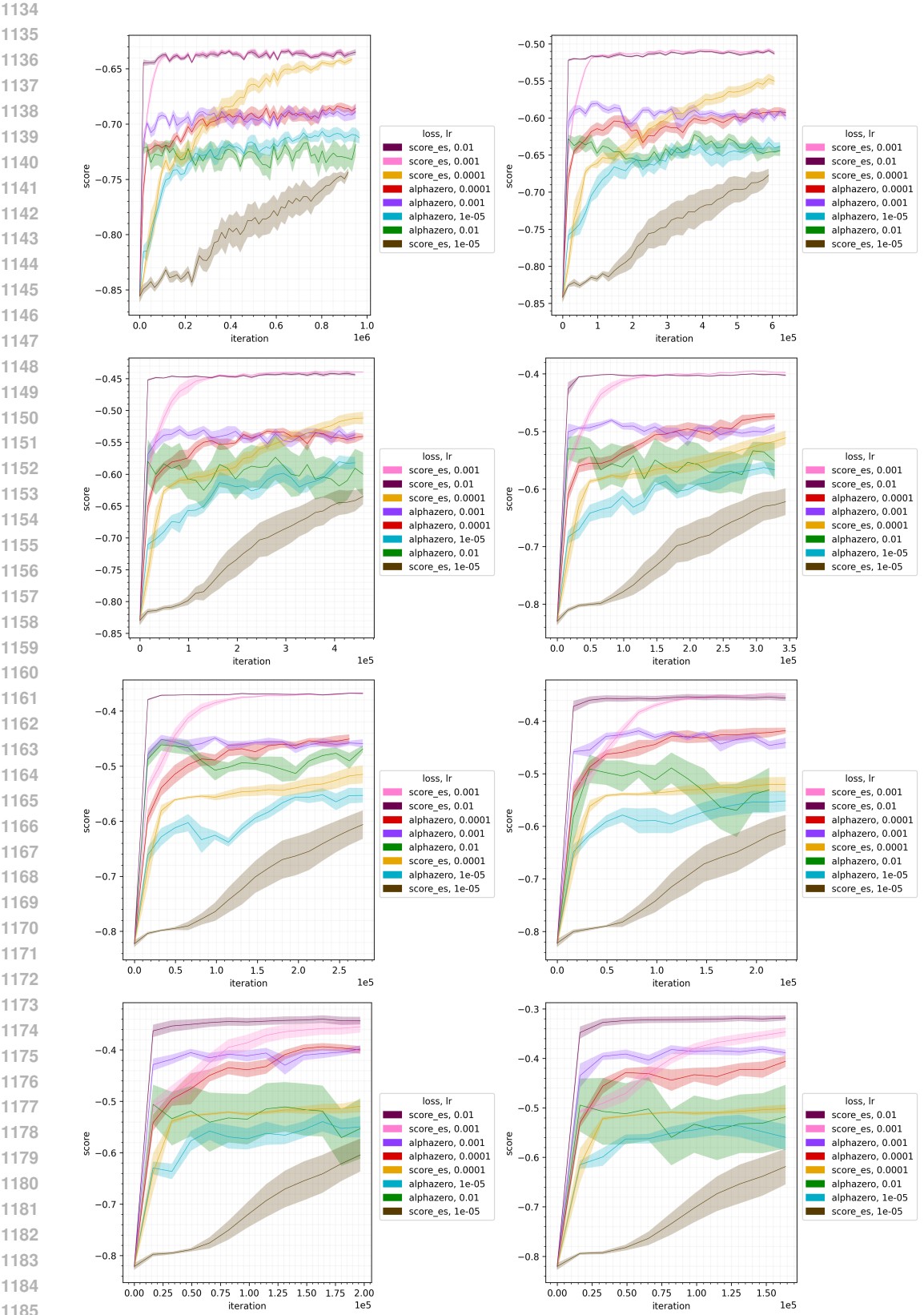

Figure 12: TSP with 8, 12, 16, 20, 24, 28, 32, and 36 points (left to right, top to bottom).

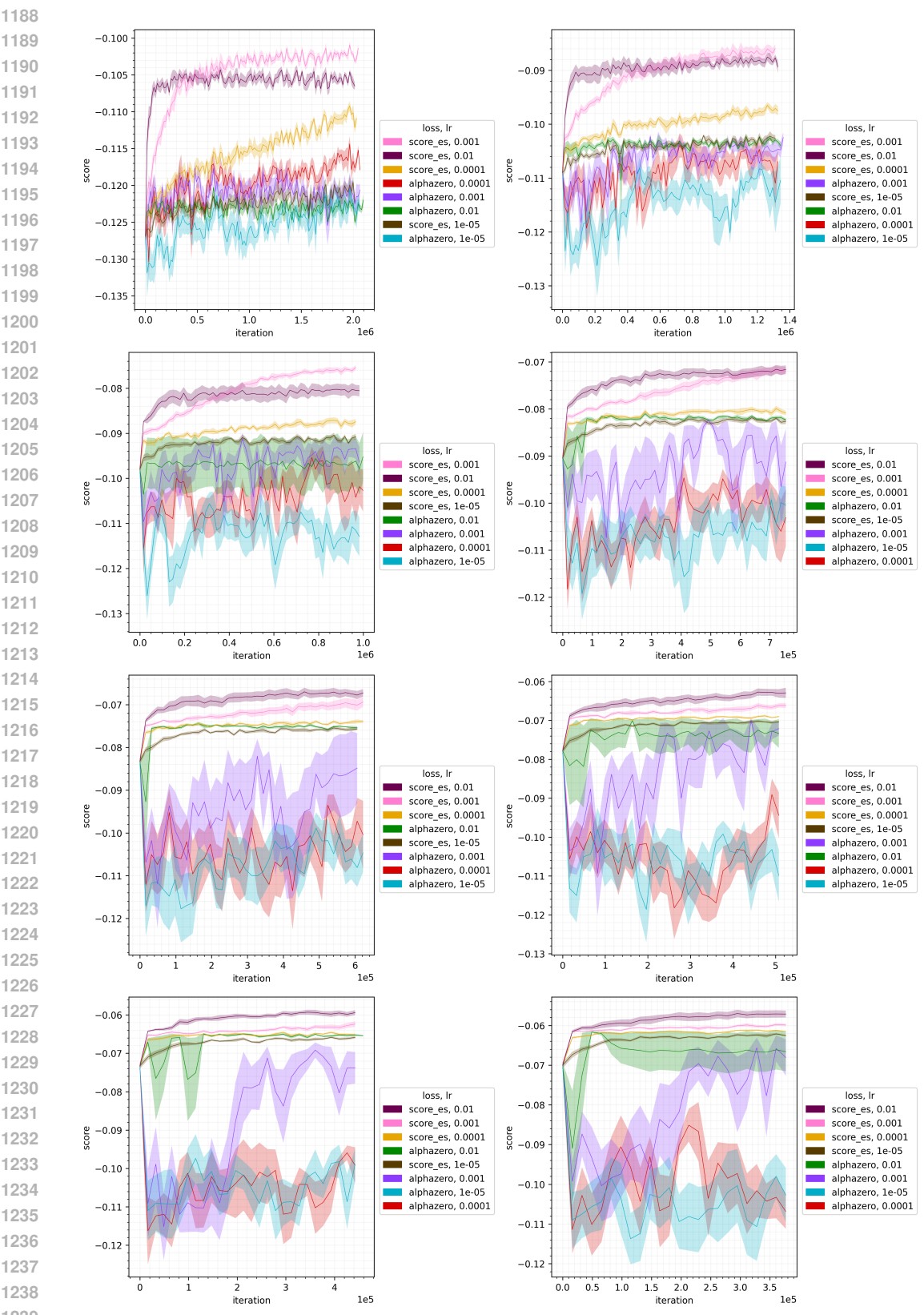

Figure 13: VKCP with 8, 12, 16, 20, 24, 28, 32, and 36 points (left to right, top to bottom). The size of the choice set is half the number of points.

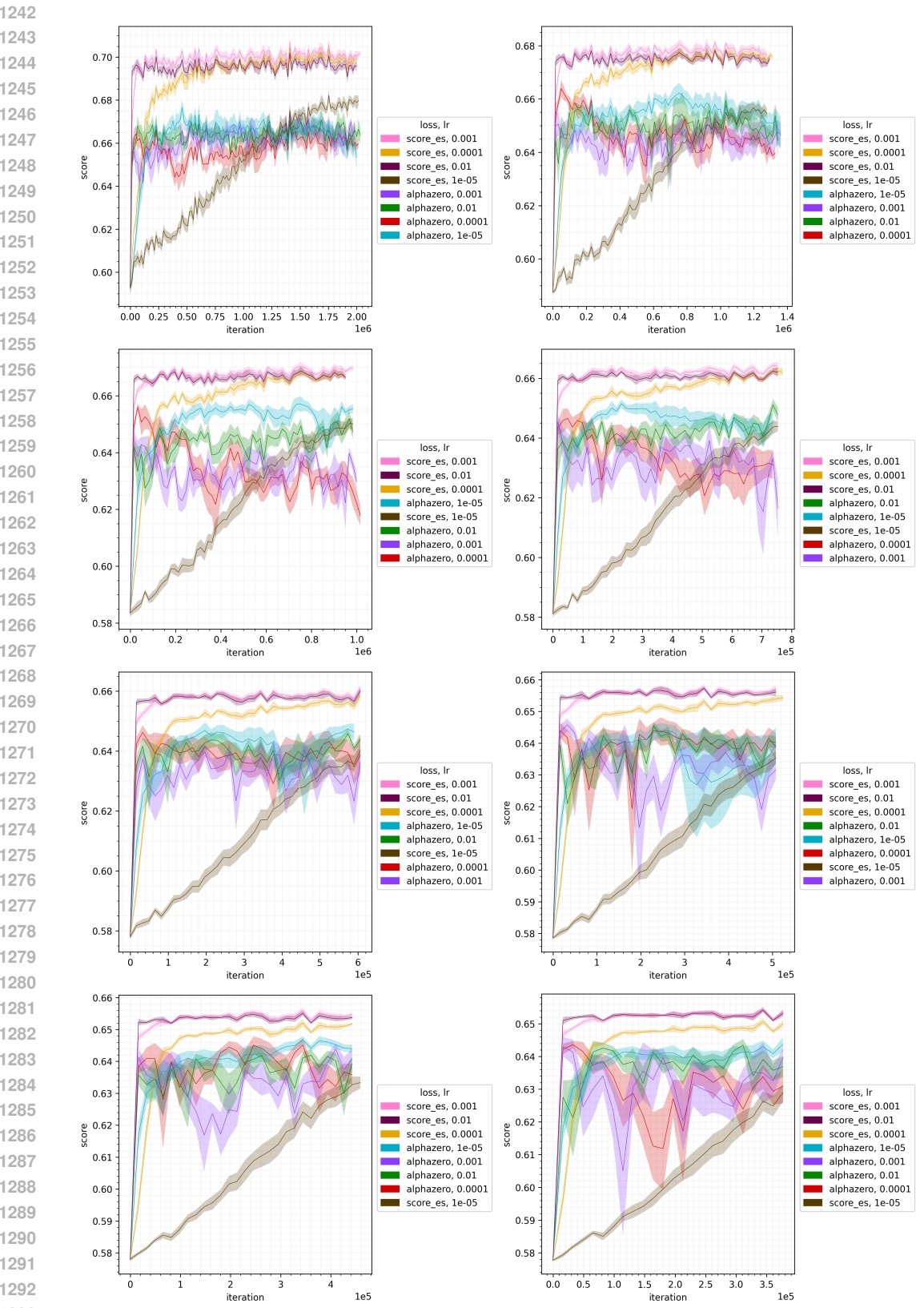

Figure 14: MDP with 8, 12, 16, 20, 24, 28, 32, and 36 points (left to right, top to bottom). The size of the choice set is half the number of points.

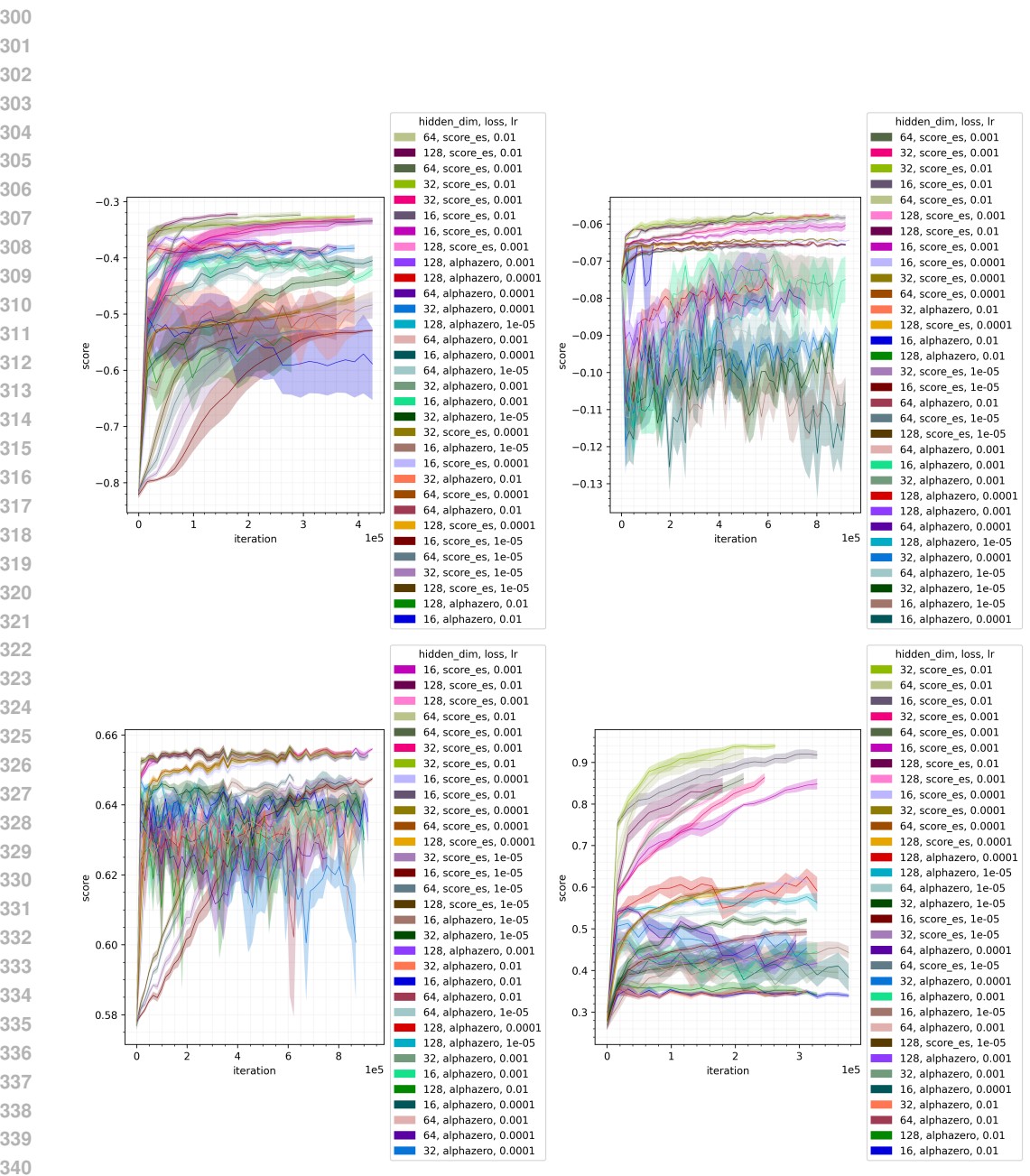

Figure 15: Performance comparison for different network sizes. Left to right, top to bottom: TSP, VKCP, MDP, and Navigation.

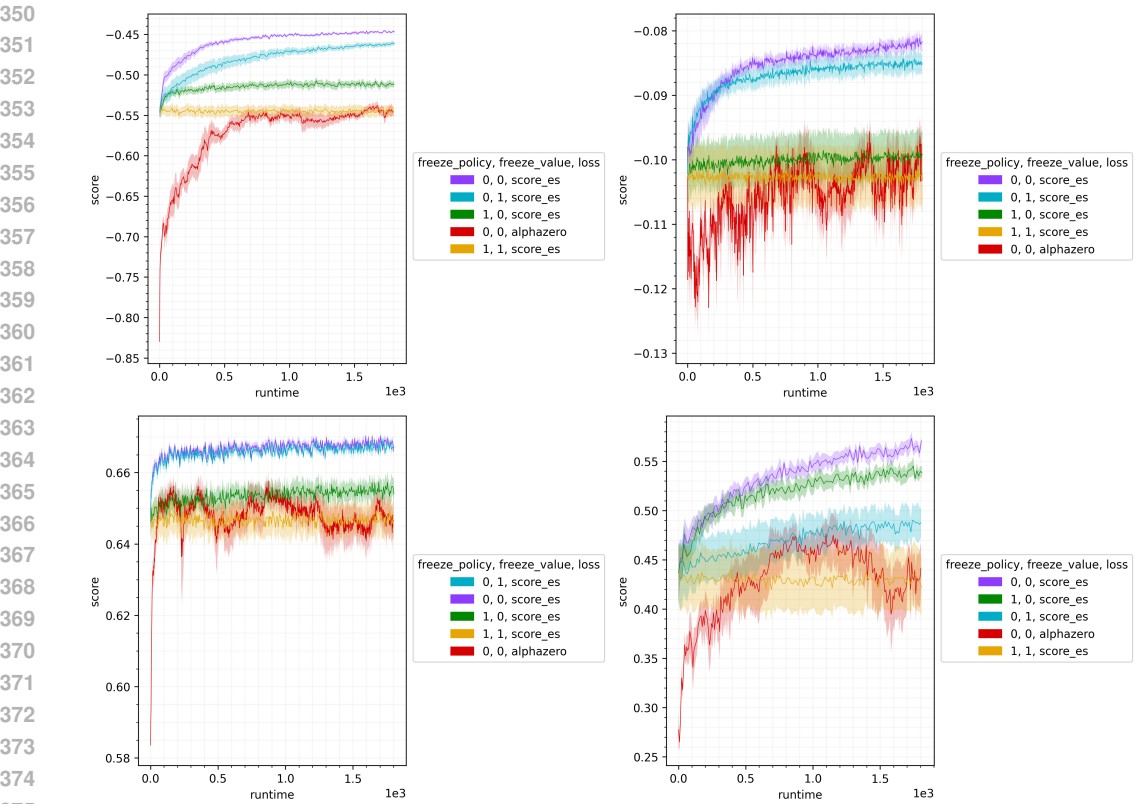

Figure 16: Ablation. Left to right, top to bottom: TSP, VKCP, MDP, and Navigation.

# E ABLATION

To further investigate where the advantage of AlphaZeroES over AlphaZero comes from, and whether most of the improvement comes from a better value or policy output, we conducted an ablation study as follows. First, we train a combined policy/value network under the standard AlphaZero loss, as described in §4 and §5. Second, we create two copies of this network and use only the value output of one (henceforth, we call it the *value* network) and the policy output of the other (henceforth, we call it the *policy* network). We do this so that we can further train the value and policy outputs separately, starting from the parameters obtained by vanilla AlphaZero. Third, we *freeze* the value network (or policy network) and train *only* the policy network (or value network) under ES.

Results are shown in Figure 16. The original AlphaZero baseline is labeled with `loss=alphazero`. The subsequent training runs, which start from the final parameters of this baseline, are labeled with `loss=score_es`. The label `freeze_policy` denotes whether the policy network is frozen. The label `freeze_value` denotes whether the value network is frozen. As expected, allowing either (or both) of these to be further trained under ES improves performance over the AlphaZero baseline. Furthermore, allowing *both* of them to be trained yields maximum performance. In some environments, namely TSP, VKCP, and MDP, freezing only the value network outperforms freezing only the policy network, suggesting that improving the policy output is more important. In other environments, namely Navigation, freezing only the policy network outperforms freezing only the value network, suggesting that improving the value output is more important. Thus, interestingly, where most of the improvement of AlphaZeroES over vanilla AlphaZero comes from—a better value output or a better policy output—is environment-dependent.

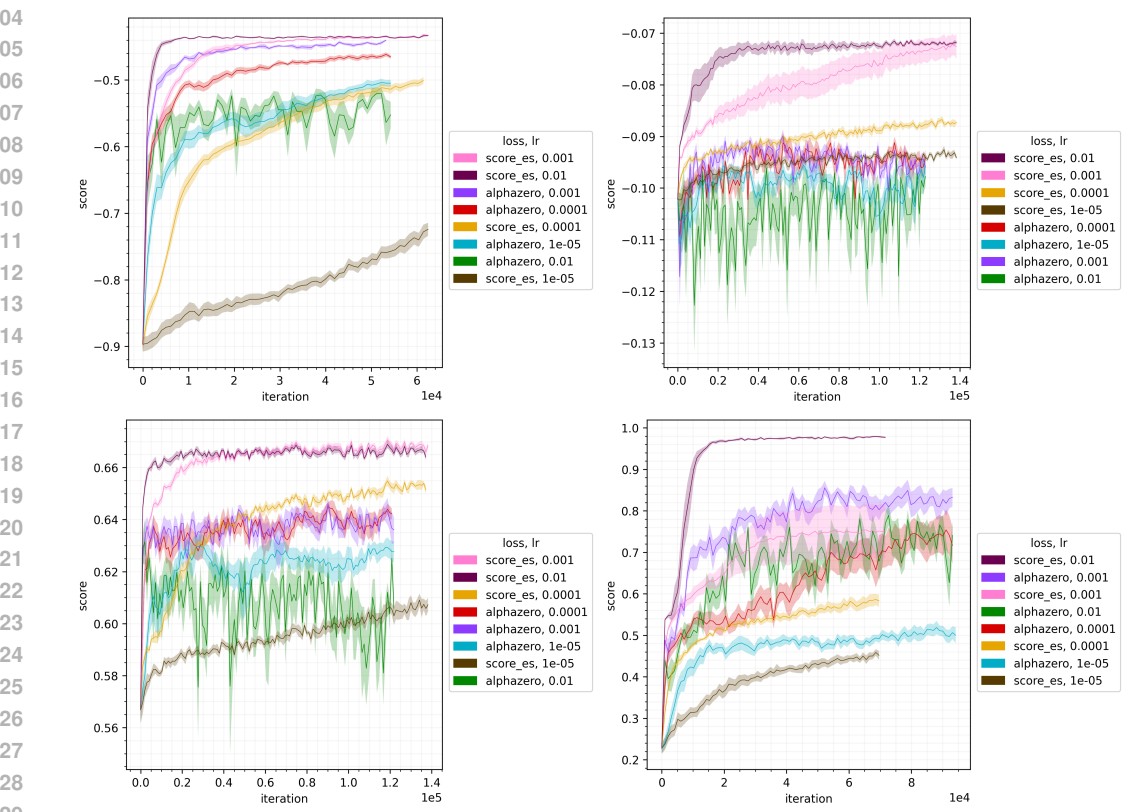

Figure 17: Performance under the attention-based architecture on TSP, VKCP, MDP, and Navigation.

## F  ARCHITECTURE COMPARISON

To check whether our approach generalizes to architectures rather than DeepSets (Zaheer et al., 2017), we run experiments with a different architecture, namely one based on neural attention (Vaswani et al., 2017). A theoretical comparison of these two architectures can be found in Wagstaff et al. (2022). Our architecture starts by applying an affine layer mapping the multiset of inputs to a multiset of hidden vectors. Then, we apply a sequence of $D$ attention blocks, where $D$ is a depth hyperparameter. (We use $D = 2$.) Each such block is a parallel attention block, as described in Zhao et al. (2019). It applies layer normalization (Ba et al., 2016), followed by a parallel application of (1) a pointwise feedforward multilayer perceptron with a single hidden layer and (2) a multi-head attention module (Vaswani et al., 2017). These two outputs are then combined with a skip connection from the input to the block, via simple addition. For reduction, we apply a many-to-one multi-head attention module on a learned readout vector initialized with random normal entries. After that, we apply the ReLU activation function followed by an affine layer. Results are shown in Figure 17. Our method, AlphaZeroES, continues to outperform AlphaZero on the new architecture.

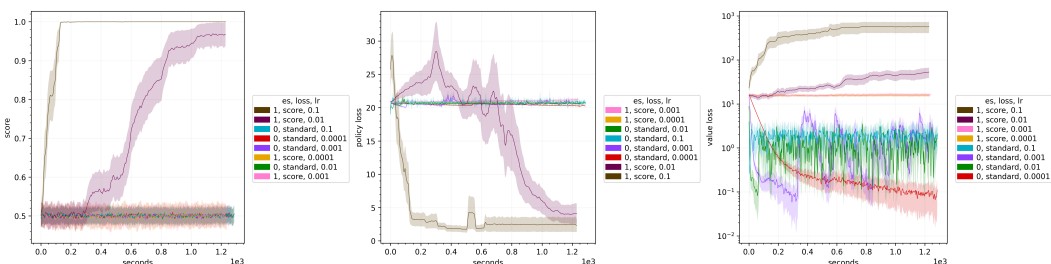

Figure 18: XOR environment metrics.

# G    FAILURE MODES FOR ALPHAZERO

In this section, we give concrete examples of *simple* environments where AlphaZero fails while AlphaZeroES succeeds.

## G.1    XOR ENVIRONMENT

Consider the following environment. A state is a triple $(b, c, t)$ where $b, c \in \{0, 1\}$ are bits and $t \in \mathbb{N}$ is the timestep. At the beginning of an episode, $b \in \{0, 1\}$ is sampled uniformly at random, $c = b$, and $t = 0$. An action is a bit $a \in \{0, 1\}$.

Letting $a$ be the current action, the transition function yields $(b, c', t + 1)$, where $c' = b \oplus a$ if $t = 0$ and $c' = c$ otherwise. In other words, $c = b \oplus a_0$ for the remainder of the episode, where $a_0$ is the initial action. At the end of the episode, the reward is $b \oplus c = b \oplus (b \oplus a_0) = a_0$. Therefore, after the initial step, the value of state $(b, c, t)$ is just $a_0$.

Therefore, this environment has an optimal policy that is very simple: always play $a = 1$. This constant policy should be easily discoverable by optimizing episode score via ES.

Suppose that we use AlphaZero with a linear function approximator for its prediction network. At the initial timestep, MCTS inspects the two successor states $(b, b, 1)$ and $(b, b \oplus 1, 1)$, and potentially their descendants, to decide which action to play. However, with a linear function approximator, AlphaZero's prediction network is unable to extract the key information $b \oplus c = b \oplus (b \oplus a_0) = a_0$, which determines the value of the state being examined.

Therefore, when AlphaZero is trained with the standard planning loss, it has no way to determine which action it should take at the initial timestep. (Provided that the episode is long enough that MCTS does not expand all the way to the terminal nodes.) On the other hand, AlphaZeroES can simply learn to always put all of the predicted prior probability on $a = 1$, which causes it to always be chosen by MCTS. Thus, we predict that AlphaZero consistently fails to learn any useful policy in this environment, while AlphaZeroES does.

In practice, we observe that this is the case. We set the number of timesteps to 32 and deployed each agent. We use only a linear (or more precisely, affine) layer for the AlphaZero prediction network, directly mapping the state to a value scalar and logits vector. Other hyperparameters are the same as in the rest of the experiments. Results are shown in Figure 18. As expected, AlphaZero fails to learn any useful policy, while AlphaZeroES learns the optimal policy.

## G.2    ENCRYPTED ENVIRONMENT

Consider the environment. Suppose that the states of the environment are "encrypted" counters. In any state, action $A$ decrypts the counter with a secret key, *increments* it, and re-encrypts it. In contrast, action $B$ does nothing. At the end of an episode, the agent receives the value of the counter. The optimal policy is very simple: always choose $A$. But learning a good value function is nearly impossible from the perspective of the agent, given that it is unable to "decrypt" states. While this example may seem extreme, given its reliance on cryptography, it is an illustrative analogy: an environment can look "encrypted" from the perspective of an agent that is not sophisticated enough (at least at the beginning of training) to "understand" what the states mean.

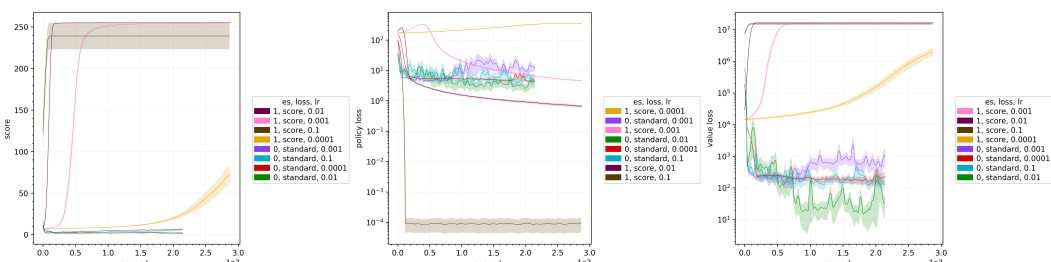

Figure 19: Encrypted environment metrics.

We implement a simple example of such an environment. For $n \in \mathbb{N}$, let $[n] = \{0, \ldots, n-1\}$. The environment's encryption function is simply a permutation $e : [256] \rightarrow [256]$. We sample this permutation uniformly at random from the set of all permutations. Likewise, the environment's decryption function is the inverse permutation $e^{-1}$.

Each state is a pair $(c, t)$, where $c \in [256]$ is the encrypted counter and $t \in [256]$ is the timestep. Given such a state, the agent observes the 8 bits of $c$, concatenated with $t/255$. The initial state is $(e(0), 0)$. Given action $a \in \{0, 1\}$, state $(c, t)$ is mapped to $(e(e^{-1}(c) + a), t + 1)$. The environment terminates when $t = 255$, and the reward is $e^{-1}(c)$.

Results are shown in Figure 19. As expected, AlphaZeroES easily learns the trivial optimal policy, while AlphaZero struggles to learn. This is because AlphaZero essentially needs to learn a big lookup table that maps each arbitrary 8-bit pattern to an arbitrary value.

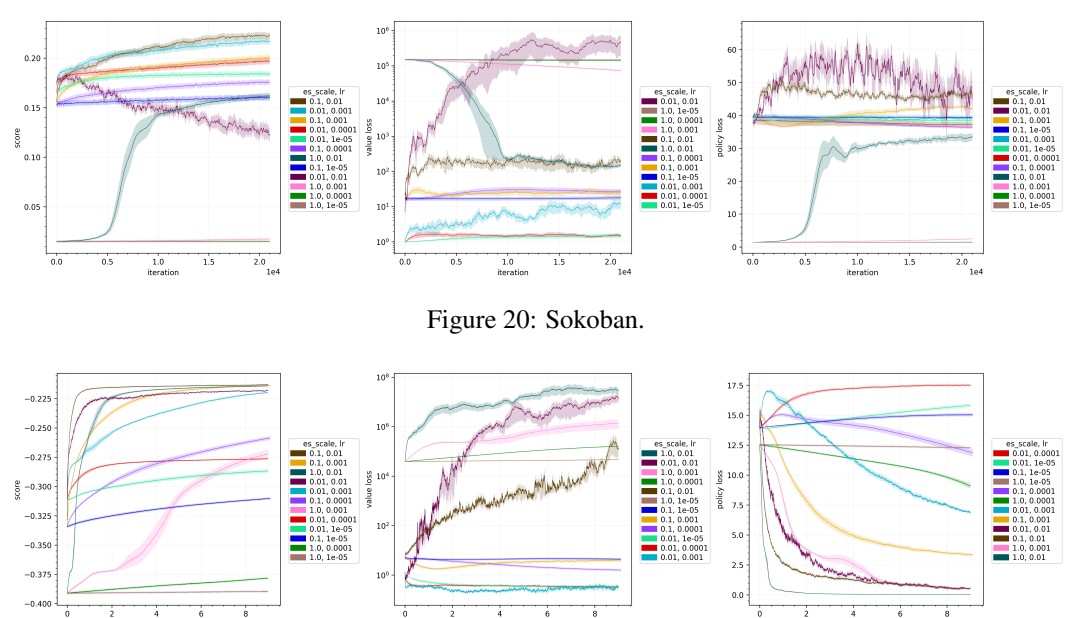

Figure 20: Sokoban.

Figure 21: TSP.

# H VARYING THE PERTURBATION SCALE

In this section, we explore what happens with different perturbation scales for AlphaZeroES. Results are shown in Figures 20–23. The results are qualitatively similar across different scales.

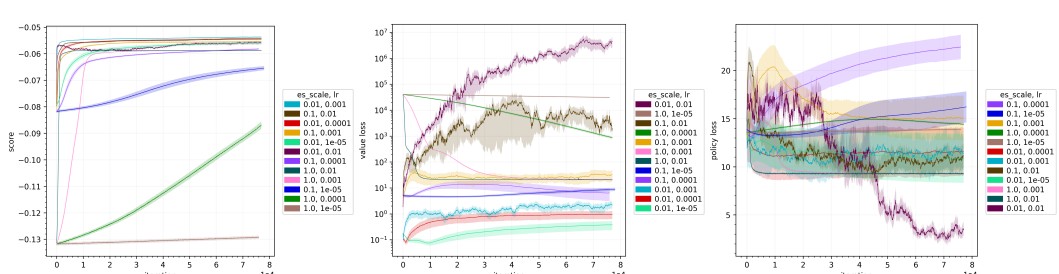

Figure 22: VKCP.

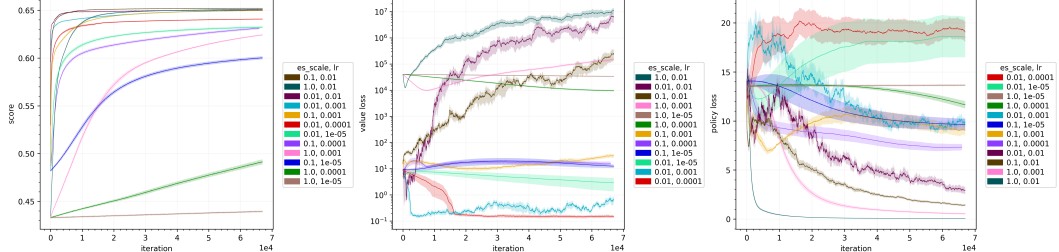

Figure 23: MDP.

# I CODE

The following is an implementation of our method in the Python programming language (Van Rossum and Drake Jr., 1995). The libraries used here are described in §5 of the paper.

File `rl_utils.py`:

```python
import jax
from jax import lax, random
from jax import numpy as jnp

def get_returns(episode):
    def f(carry, reward_discount):
        reward, discount = reward_discount
        new_carry = reward + discount * carry
        return new_carry, new_carry

    rewards = episode["reward"]
    discounts = episode["discount"]
    init = jnp.zeros(rewards.shape[1:])
    xs = rewards, discounts
    _, returns = lax.scan(f, init, xs, unroll=True, reverse=True)
    return returns

def get_reach_probs(episode):
    discounts = episode["discount"]
    reach_probs = jnp.cumprod(discounts[:-1])
    reach_probs = jnp.insert(reach_probs, 0, 1)
    return reach_probs

def get_score(episode):
    reach_probs = get_reach_probs(episode)
    return reach_probs @ episode["reward"]

def sample_episode(env, agent, params, key, unroll=1):
    def step(state_memory, key):
        state, memory = state_memory
        action, new_memory, agent_extra = agent.apply(params, state, key,
          ↪  memory)
        reward, discount, new_state = env.step(state, action)
        return (new_state, new_memory), {
            "state": state,
            "action": action,
            "agent_extra": agent_extra,
            "reward": reward,
            "discount": discount,
            "memory": memory,
        }

    key, subkey = random.split(key)
    state = env.init(subkey)

    key, subkey = random.split(key)
    memory = agent.init_memory(subkey)

    keys = random.split(key, env.max_steps())
    (state, memory), episode = lax.scan(step, (state, memory), keys,
      ↪  unroll=unroll)

    episode["state"] = jax.tree.map(
        lambda xs, x: jnp.concatenate([xs, x[None]]),
```

```
        episode["state"],                                                    57
        state,                                                               58
    )                                                                        59
                                                                             60
    episode["memory"] = jax.tree.map(                                        61
        lambda xs, x: jnp.concatenate([xs, x[None]]),                        62
        episode["memory"],                                                   63
        memory,                                                              64
    )                                                                        65
                                                                             66
    return episode                                                           67
                                                                             68
                                                                             69
def get_num_actions(env):                                                    70
    key = random.key(0)                                                      71
    state = env.init(key)                                                    72
    space = env.action_space(state)                                          73
    return space.mask.size                                                   74
```

File rl_losses.py:

```
import optax                                                                  1
from jax import lax, nn                                                       2
from jax import numpy as jnp                                                  3
                                                                             4
from lib.rl_utils import get_reach_probs, get_returns                        5
                                                                             6
                                                                             7
def mcts_action_loss(episode):                                               8
    predictions = episode["agent_extra"]["mcts_action_prediction"]           9
    targets = episode["agent_extra"]["mcts_action_target"]                  10
    mask = episode["agent_extra"]["mcts_action_mask"]                       11
    losses = optax.kl_divergence(                                           12
        nn.log_softmax(predictions, where=mask),                           13
        lax.stop_gradient(targets),                                        14
        where=mask,                                                        15
    )                                                                      16
    return get_reach_probs(episode) @ losses                               17
                                                                           18
                                                                           19
def mcts_value_loss_mc(episode):                                           20
    """Monte Carlo."""                                                     21
    predictions = episode["agent_extra"]["mcts_value_prediction"]          22
    targets = get_returns(episode)                                         23
    losses = optax.squared_error(                                          24
        predictions,                                                       25
        lax.stop_gradient(targets),                                        26
    )                                                                      27
    return get_reach_probs(episode) @ losses                               28
                                                                           29
                                                                           30
def mcts_value_loss_dp(episode):                                           31
    """Dynamic programming or self-bootstrapping."""                       32
    predictions = episode["agent_extra"]["mcts_value_prediction"]          33
    targets = episode["agent_extra"]["mcts_value_target"]                  34
    losses = optax.squared_error(                                          35
        predictions,                                                       36
        lax.stop_gradient(targets),                                        37
    )                                                                      38
    return get_reach_probs(episode) @ losses                               39
                                                                           40
                                                                           41
def alphazero_loss(episode):                                               42
    value_loss = mcts_value_loss_mc(episode)                               43
```

```python
    action_loss = mcts_action_loss(episode)                    44
    loss = value_loss + action_loss                            45
    metrics = {                                                46
        "value_loss": value_loss,                              47
        "action_loss": action_loss,                            48
        "loss": loss,                                          49
                                                               50
    }                                                          51
    return loss, metrics                                       52
                                                               53
                                                               54
def mcts_consistency_loss(episode):                            55
    value_loss = mcts_value_loss_dp(episode)                   56
    action_loss = mcts_action_loss(episode)                    57
    loss = value_loss + action_loss                            58
    metrics = {                                                59
        "value_loss": value_loss,                              60
        "action_loss": action_loss,                            61
        "loss": loss,                                          62
    }                                                          63
    return loss, metrics
```

File `mcts.py`:

```python
import jax                                                     1
import mctx                                                    2
from jax import lax, nn                                        3
from jax import numpy as jnp                                   4
                                                               5
                                                               6
def gumbel_muzero(                                             7
    state,                                                     8
    prediction_fn,                                             9
    step_fn,                                                   10
    action_mask,                                               11
    budget,                                                    12
    key,                                                       13
    algorithm="gumbel_muzero",                                 14
    **kwargs,                                                  15
):                                                             16
    def root_fn(state):                                        17
        value, logits = prediction_fn(state)                   18
        return mctx.RootFnOutput(                              19
            prior_logits=logits,  # type: ignore               20
            value=value,  # type: ignore                       21
            embedding=state,  # type: ignore                   22
        )                                                      23
                                                               24
    def recurrent_fn(params, key, action, state):             25
        reward, discount, new_state = step_fn(state, action)   26
        value, logits = prediction_fn(new_state)               27
        output = mctx.RecurrentFnOutput(                       28
            reward=reward,  # type: ignore                     29
            discount=discount,  # type: ignore                 30
            prior_logits=logits,  # type: ignore               31
            value=value,  # type: ignore                       32
        )                                                      33
        return output, new_state                               34
                                                               35
    algorithm_fn = {                                           36
        "gumbel_muzero": mctx.gumbel_muzero_policy,            37
        "muzero": mctx.muzero_policy,                          38
    }[algorithm]                                               39
                                                               40
    root = root_fn(state)                                      41
```

```
outputs = algorithm_fn(                                              42
    params=(),                                                       43
    rng_key=key,                                                     44
    root=jax.tree.map(lambda x: jnp.expand_dims(x, 0), root),        45
    recurrent_fn=jax.vmap(recurrent_fn, [None, None, 0, 0]),         46
    num_simulations=budget + 1,                                      47
    invalid_actions=jax.tree.map(lambda x: jnp.expand_dims(~x, 0),   48
    ↪ action_mask),
    **kwargs,                                                        49
)                                                                    50
summary = jax.tree.map(lambda x: x[0], outputs.search_tree.summary()) 51
output = jax.tree.map(lambda x: x[0], outputs)                       52
return {                                                             53
    "action": output.action,                                        54
    "action_onehot": nn.one_hot(output.action, output.action_weights. 55
    ↪ size),
    "action_weights": lax.stop_gradient(output.action_weights),     56
    "root_value": root.value,                                       57
    "root_logits": root.prior_logits,                               58
    "root_state": state,                                            59
    "search_tree": lax.stop_gradient(output.search_tree),           60
    "visit_counts": summary.visit_counts,                           61
    "visit_probs": summary.visit_probs,                             62
    "value": lax.stop_gradient(summary.value),                      63
    "qvalues": summary.qvalues,                                     64
    "action_mask": action_mask,                                     65
}                                                                    66
```

File alphazero.py:

```
from functools import partial                                        1
                                                                     2
from lib import mcts                                                 3
                                                                     4
                                                                     5
class AlphaZero:                                                     6
                                                                     7
    def __init__(self, env, pred_fn, budget):                       8
        self.env = env                                               9
        self.pred_fn = pred_fn                                      10
        self.budget = budget                                       11
                                                                    12
    def init(self, params_key, state, key, memory):                13
        return self.pred_fn.init(params_key, state)                14
                                                                    15
    def init_memory(self, key):                                    16
        return None                                                17
                                                                    18
    def apply(self, params, state, key, memory):                   19
        space = self.env.action_space(state)                       20
        output = mcts.gumbel_muzero(                                21
            state=state,                                           22
            prediction_fn=partial(self.pred_fn.apply, params),     23
            step_fn=self.env.step,                                 24
            budget=self.budget,                                    25
            key=key,                                               26
            action_mask=space.mask,                                27
        )                                                          28
        return (                                                   29
            output["action"],                                     30
            memory,                                                31
            {                                                      32
                "search_tree": output["search_tree"],             33
                "mcts_value_prediction": output["root_value"],    34
```

```
                    "mcts_value_target": output["value"],                    35
                    "mcts_action_prediction": output["root_logits"],         36
                    "mcts_action_target": output["action_weights"],          37
                    "mcts_action_mask": space.mask,                          38
                },                                                           39
            )                                                               40
```

File predictors.py:

```
import argparse                                                              1
                                                                            2
from flax import linen as nn                                                 3
from jax import numpy as jnp                                                 4
                                                                            5
from lib import envs, rl_utils                                               6
                                                                            7
                                                                            8
class DensePredictor(nn.Module):                                            9
    args: argparse.Namespace                                                10
    env: envs.Env                                                           11
                                                                            12
    @nn.compact                                                             13
    def __call__(self, state):                                             14
        x = self.env.observation_vector(state)                             15
        x = nn.Dense(self.args.hidden_dim)(x)                              16
        x = nn.relu(x)                                                     17
                                                                            18
        logits = nn.Dense(rl_utils.get_num_actions(self.env))(x)           19
                                                                            20
        if hasattr(self.env, "players"):                                   21
            values = nn.Dense(self.env.players)(x)                         22
            return values, logits                                          23
        else:                                                              24
            (value,) = nn.Dense(1)(x)                                      25
            return value, logits                                           26
                                                                            27
                                                                            28
class DeepSetsPredictor(nn.Module):                                        29
    args: argparse.Namespace                                                30
    env: envs.Env                                                          31
                                                                            32
    @nn.compact                                                            33
    def __call__(self, state):                                             34
        x, mask = self.env.observation_multiset(state)                     35
        if mask is None:                                                   36
            mask = jnp.ones(x.shape[0], bool)                              37
                                                                            38
        for _ in range(self.args.depth):                                  39
            x_skip = x                                                     40
            x = nn.Dense(self.args.hidden_dim)(x)                          41
            x = nn.relu(x)                                                 42
            x1 = nn.Dense(self.args.hidden_dim)(x.sum(0, where=mask[...,   43
            ↪ None]))
            x1 /= 1 + mask.sum(0)[..., None]                               44
            x2 = nn.Dense(self.args.hidden_dim, use_bias=False)(x)         45
            x = x1 + x2                                                    46
            x = nn.relu(x)                                                 47
            if x_skip.shape == x.shape:                                    48
                x += x_skip                                                49
                                                                            50
        match self.env:                                                    51
            case (                                                         52
                envs.EuclideanTSP()                                        53
                | envs.Knapsack()                                          54
```

```
                     | envs.EuclideanFLP()                                  55
                     | envs.SubsetSum()                                     56
                     | envs.MaximumDiversityProblem()                       57
                     | envs.MaxLengthTSP()                                  58
                 ):                                                         59
                     logits = nn.Dense(1)(x)[..., 0]                        60
                 case envs.Sokoban() | envs.Reach():                       61
                     y = x.mean(0, where=mask[..., None])                   62
                     logits = nn.Dense(rl_utils.get_num_actions(self.env))(y)  63
                 case _:                                                    64
                     breakpoint()                                           65
                     raise NotImplementedError(self.env)                    66
                                                                           67
        x = x.mean(0, where=mask[..., None])                               68
                                                                           69
        if hasattr(self.env, "players"):                                  70
            values = nn.Dense(self.env.players)(x)                         71
            return values, logits                                          72
        else:                                                              73
            (value,) = nn.Dense(1)(x)                                      74
            return value, logits                                          75
                                                                           76
                                                                           77
class AttentionPredictor(nn.Module):                                       78
    args: argparse.Namespace                                               79
    env: envs.Env                                                          80
                                                                           81
    @nn.compact                                                            82
    def __call__(self, state):                                            83
        x, mask = self.env.observation_multiset(state)                    84
        if mask is None:                                                  85
            mask = jnp.ones(x.shape[0], bool)                             86
                                                                           87
        x = nn.Dense(self.args.hidden_dim)(x)                             88
                                                                           89
        for _ in range(self.args.depth):                                 90
            x_norm = nn.LayerNorm(use_bias=False, use_scale=False)(x)      91
                                                                           92
            y = nn.Dense(self.args.hidden_dim)(x_norm)                     93
            y = nn.relu(y)                                                94
            y = nn.Dense(self.args.hidden_dim, kernel_init=nn.            95
             ↪ initializers.zeros)(y)
                                                                           96
            z = nn.MultiHeadAttention(self.args.heads)(x_norm, mask=mask)  97
                                                                           98
            x += y + z                                                    99
                                                                          100
        match self.env:                                                  101
            case (                                                       102
                envs.EuclideanTSP()                                      103
                | envs.Knapsack()                                        104
                | envs.EuclideanFLP()                                    105
                | envs.SubsetSum()                                       106
                | envs.MaximumDiversityProblem()                         107
                | envs.MaxLengthTSP()                                    108
            ):                                                          109
                logits = nn.Dense(1)(x)[..., 0]                         110
            case envs.Sokoban() | envs.Reach():                        111
                y = x.mean(0, where=mask[..., None])                    112
                logits = nn.Dense(rl_utils.get_num_actions(self.env))(y)  113
            case _:                                                     114
                breakpoint()                                            115
                raise NotImplementedError(self.env)                     116
                                                                        117
        readout = self.param(                                          118
```

```python
            "readout", nn.initializers.normal(1), [self.args.hidden_dim]
        )
        x = nn.MultiHeadAttention(self.args.heads)(readout[None], x, mask
          ↪ =mask).squeeze(
            0
        )

        if hasattr(self.env, "players"):
            values = nn.Dense(self.env.players)(x)
            return values, logits
        else:
            (value,) = nn.Dense(1)(x)
            return value, logits

class MixedPredictor(nn.Module):
    value: nn.Module
    policy: nn.Module

    @nn.compact
    def __call__(self, state):
        value, _ = self.value(state)
        _, logits = self.policy(state)
        return value, logits
```

File pseudogradient.py:

```python
from functools import partial

import jax
import optax
from jax import lax, random
from jax import numpy as jnp
from jax.scipy import stats
from optax import tree_utils as otu

class Normal:
    def __init__(self, loc, scale):
        self.loc = loc
        self.scale = scale

    def sample(self, key):
        z = otu.tree_random_like(key, self.loc)
        return jax.tree.map(lambda l, z: l + self.scale * z, self.loc, z)

    def sample_antithetic(self, key):
        z = otu.tree_random_like(key, self.loc)
        return jax.tree.map(
            lambda l, z: l + self.scale * jnp.stack([z, -z]),
            self.loc,
            z,
        )

    def logpdf(self, x):
        logpdfs = jax.tree.map(
            lambda l, x: stats.norm.logpdf(x, l, self.scale),
            self.loc,
            x,
        )
        return otu.tree_sum(logpdfs)

def smoothe(scale, distribution="normal"):
```

```
    match distribution:
        case "normal":
            distribution_cls = Normal
        case _:
            raise NotImplementedError

    def g(f, x, key):
        dist = distribution_cls(x, scale)

        key, subkey = random.split(key)
        samples = lax.stop_gradient(dist.sample_antithetic(subkey))

        outputs = jax.vmap(f, [0, None], axis_size=2)(samples, key)

        log_probs = jax.vmap(dist.logpdf, axis_size=2)(samples)
        assert log_probs.ndim == 1

        ones = jnp.exp(log_probs - lax.stop_gradient(log_probs))
        ones /= ones.size

        return jax.tree.map(lambda outputs: ones @ outputs, outputs)

    return lambda f: partial(g, f)
```

