# OpenReview forum: "AlphaZeroES: Direct Score Maximization Can Outperform Planning Loss Minimization in Single-Agent Settings"
_ICLR.cc/2026/Conference — ICLR 2026 Conference Withdrawn Submission_

### Official Review · Reviewer_ygAX · 2025-10-25

**Soundness:** 2
**Presentation:** 2
**Contribution:** 2
**Rating:** 4
**Confidence:** 4

**Summary:**

This paper proposes AlphaZeroES, a modification to the AlphaZero algorithm for single-agent settings. The core idea is to replace AlphaZero's standard planning loss—which minimizes the difference between network predictions and MCTS search results (for policy) and episode returns (for value) —with a new objective that *directly* maximizes the total episode score. Because the MCTS component is non-differentiable, the authors employ Evolution Strategies (ES) as a zeroth-order, black-box optimizer to train the neural network parameters. The method is evaluated on five single-agent environments (Navigation, Sokoban, TSP, VKCP, and MDP), where the authors claim that AlphaZeroES "dramatically outperforms" the standard AlphaZero baseline while using the same MCTS algorithm and network architecture.

**Strengths:**

The paper's primary strength is the question it poses: challenging the optimality of the standard AlphaZero planning loss. This is a fundamental and worthwhile question to investigate.

The paper's observation that maximizing the score via ES does not correlate with minimizing the standard planning losses (and can even be anti-correlated) is an interesting finding.

The ablation study in Appendix D, which attempts to separate the contributions of the policy and value network optimization, is a good addition and provides some insight, showing the source of improvement is environment-dependent.

**Weaknesses:**

My main concerns are as follows:

1.  **Lack of Intuition and Analysis:** The paper provides no satisfying explanation for *why* its method works. In fact, the results in Figures 2, 3, 5, etc., show that the value and policy losses for AlphaZeroES often increase or stagnate while the score improves. The paper even states "a definitive explanation... is beyond the scope of this paper". This is not acceptable. If the network's value/policy heads are producing "worse" predictions (according to the standard loss), how is the MCTS using these heuristics to produce a *better* overall policy? What is being learned? Without this analysis, the paper is just a collection of puzzling empirical results.

2.  **Questionable Sample Efficiency:** Evolution Strategies are zeroth-order methods and are notoriously sample-inefficient, especially in high-dimensional parameter spaces (like neural networks). Yet, this paper claims comparable or better performance within the same training time and number of episodes as gradient-based AlphaZero. This is an extraordinary claim. It suggests that ES is *more* sample-efficient than backpropagation for this complex planning problem, which contradicts a large body of literature. This result is highly questionable and may be an artifact of poorly tuned baselines or very simple environments.

3.  **Vague Methodology & Reproducibility:** As mentioned in the Presentation section, the paper lacks a clear, high-level pseudocode for the AlphaZeroES training loop. Section 4.3 just describes ES, not its integration. This, combined with the lack of a complete, runnable code repository, makes the results difficult to trust or replicate.

4.  **Poor Scalability and Simple Environments:** The experiments are conducted on "toy" problems. A 10x10 grid or a 20-city TSP is not a convincing demonstration of a method intended to improve *AlphaZero*. AlphaZero's fame comes from its ability to master extremely complex domains like Go or chess. The scalability experiments in Appendix C only test up to 36 nodes and 128 hidden dimensions. This is insufficient. It is highly likely that a black-box ES approach will fail to scale to the millions of parameters and vast search spaces where gradient-based AlphaZero excels.

**Questions:**

1.  Could the authors provide a more concrete analysis of the learning dynamics? If the value loss is high (e.g., Fig 2), does this mean the MCTS effectively learns to ignore the value head's output and relies purely on rollouts? What do the learned policy/value predictions actually look like? How can the search be effective if its guiding heuristics are, by the paper's own metrics, not improving?

2.  Can you please comment on the surprising sample efficiency? Why would a zeroth-order method (ES) be more efficient than backpropagation here? Were you able to run the baseline AlphaZero with a fully optimized set of hyperparameters? The results are very counterintuitive.

3.  To properly test the claims of scalability, would it be possible to test this on a much larger, more standard benchmark? For example, a larger combinatorial problem (e.g., TSP with $n=100$) or a different domain entirely (like a small board game)? The current environments are too simple to support the paper's strong claims.

4.  Section 5 states that "AlphaZero and AlphaZeroES took about the same amount of time per iteration." This is a very surprising claim. A standard ES update (like OpenAI-ES) requires $N$ full episode rollouts (one for each worker/perturbation) just to compute a *single* pseudogradient $g$. In contrast, a standard AlphaZero update (or batch update) can learn from the data of a single episode (or batch of episodes) via backpropagation. Could you please clarify what "iteration" refers to in this context? Does it mean one full parameter update, or one generated episode? This claim is central to the method's viability but seems to misrepresent the computational cost of ES.

---

> ### Author Response · Authors · 2025-12-04
>
> > Lack of Intuition and Analysis
>
> We have edited the conclusion to make our explanation clearer.
>
> We have also added new sections to the appendix giving simple, concrete environments where AlphaZero fails while AlphaZeroES succeeds. These substantiate our explanation.
>
> > Questionable Sample Efficiency
>
> The citations we give in line 202 (Salimans et al. 2017) and 203 (Lenc et al. 2019) show that ES-based methods are indeed scalable. ES has even been applied successfully to LLM training: Qiu et al. *Evolution Strategies at Scale: LLM Fine-Tuning Beyond Reinforcement Learning*. 2025.
>
> Furthermore, we explicitly quote from Salimans et al. 2017 in Section C of the appendix:
>
> > The resemblance of ES to finite differences suggests the method will scale poorly with the dimension of the parameters. However, it is important to note that this does not mean that larger neural networks will perform worse than smaller networks when optimized using ES: **what matters is the difficulty, or intrinsic dimension, of the optimization problem**.
>
> Furthermore, the superiority of AZES over AZ is due not to a superiority of ES over gradient-based methods, but rather the fact that AZES *optimizes a different objective*, namely, it directly maximizes the episode score. The ES is merely a means to this end. This is explained in lines 48-50 and 186-188. AZES is not doing ES on AlphaZero’s loss, but rather changing the loss itself.
>
> > Vague Methodology & Reproducibility
>
> A step-by-step description is given in lines 207-227. Pseudo-code is given in Algorithm 2 of Salimans et al. 2017, which is cited immediately before that. We have pasted that pseudo-code there, with attribution, for clarity. We also provide code in the appendix.
>
> > This, combined with the lack of a complete, runnable code repository, makes the results difficult to trust or replicate.
>
> We have provided the code in Section F of the appendix. We also describe the libraries we use, with exact versions, in Section 5.
>
> > The experiments are conducted on "toy" problems.
>
> These are not “toy” problems. They are standard combinatorial optimization (CO) problems. They are very well known in the CO literature. We give citations supporting this in each environment’s description. We also describe the CO literature in greater detail in the “Additional Related Work” section of the appendix.
>
> > A 10x10 grid or a 20-city TSP is not a convincing demonstration of a method intended to improve AlphaZero. AlphaZero's fame comes from its ability to master extremely complex domains like Go or chess.
>
> This argument is fallacious.
> Chess takes place on an 8x8 grid, which is even smaller.
> Sokoban is a difficult planning problem. In fact, it is PSPACE-complete. This is stated in lines 303-304.
>
> > It is highly likely that a black-box ES approach will fail to scale to the millions of parameters and vast search spaces where gradient-based AlphaZero excels.
>
> There is no evidence behind this assertion, and there is evidence against it. See our response to the same point above regarding ES.
>
> > Can you please comment on the surprising sample efficiency? Why would a zeroth-order method (ES) be more efficient than backpropagation here?
>
> Backpropagation (in the style of REINFORCE or policy gradients) is simply not possible, because MCTS is not differentiable. This is explained in lines 48 and 185-188.
>
> Instead, AlphaZero and AlphaZeroES are optimizing entirely different objectives (planning loss vs. episode score). They’re not optimizing the same objective via different optimization methods. The point is that *score maximization* is more effective than planning loss minimization, not that ES is more efficient than backpropagation.
>
> > Were you able to run the baseline AlphaZero with a fully optimized set of hyperparameters?
>
> Yes. We test a range of different learning rates for each method, and do extensive hyperparameter comparisons for problem size, network size, and architecture in the appendix.
>
> > The results are very counterintuitive.
>
> It might seem that way, but we believe our hypothesis in lines 434-448 is also intuitive. AZES is directly optimizing the quantity of ultimate interest (episode score), while AZ is doing that only in a much more indirect way.
>
> The new experiments in the appendix on very simple environments give concrete examples demonstrating this hypothesis.
>
> > A standard ES update (like OpenAI-ES) requires N full episode rollouts (one for each worker/perturbation) just to compute a single pseudogradient . In contrast, a standard AlphaZero update (or batch update) can learn from the data of a single episode (or batch of episodes) via backpropagation.
>
> Both AZ and AZES use a batch of N episodes per iteration.
>
> > Could you please clarify what "iteration" refers to in this context? Does it mean one full parameter update, or one generated episode?
>
> An iteration is a parameter update. Each iteration uses N episodes. This N is the same for both methods.

---

### Official Review · Reviewer_f9hi · 2025-10-26

**Soundness:** 2
**Presentation:** 2
**Contribution:** 2
**Rating:** 2
**Confidence:** 3

**Summary:**

This paper introduces AlphaZeroES, which replaces AlphaZero's planning loss minimization with direct episode score maximization using evolution strategies (ES). The authors test this on single-agent environments including Navigation, Sokoban, TSP, VKCP, and MDP, and report that AlphaZeroES consistently outperforms standard AlphaZero while maintaining the same MCTS algorithm and neural architecture.

**Strengths:**

1. The paper poses a focused, well-motivated question about whether direct score optimization can outperform indirect planning loss minimization.
2. Comprehensive demonstration of related work is provided.
3. Limitation is honestly discussed.
4. Statistical tests are provided, including Wilcoxon, signed-rank, and paired t-tests, which show statistical significance.

**Weaknesses:**

1. The organization and presentation of the paper can be improved. It would be better put mathematical expressions separately for clear elaboration instead of using in-line mode. For Section 5 (Experiments), the authors spend a great deal of space to introduce the environments instead of discussing the quantitative results demonstrated in the figures. Moreover, for each environment, all contents are put in huge bulk of single paragraph, making it difficult to follow.
2. The paper lacks theoretical justification. In Section 6 (Discussion), the claim about "simple optimal policy but complex value function" lacks rigor and doesn't explain why this would systematically favor ES.
3. The paper claims to test "across a wide range of learning rates for fair comparison" between AlphaZero and AlphaZeroES. However, this doesn't ensure a fair comparison. Different learning rates are optimal for different objectives, making it impossible to isolate whether improvements come from the objective change or better hyperparameter selection.
4. The loss report is inconsistent. Figures show AlphaZeroES doesn't minimize value/policy losses, yet performance improves. This disconnect isn't adequately explained.
5. Sensitivity analysis is not provided regarding perturbation scale, which is set to be 0.1. The ES-specific hyperparameter selection is not explained.

**Questions:**

1. Did the authors perform a grid search or other systematic hyperparameter optimization for both methods? If so, what was the protocol?
2. How did the authors ensure the learning rate ranges tested were appropriate for each method's objective? Did the authors verify convergence for both methods at their optimal learning rates?
3. How did the authors select the perturbation scale of 0.1 for ES? What happens with different scales, such as 0.01, 0.05, 0.2, or adaptive schedules?
4. Regarding loss report, if AlphaZeroES achieves high scores without minimizing value/policy losses, what exactly has it learned? Could the authors analyze the learned representations? Could the authors plot the correlation between planning loss and episode score throughout training for both methods?
5. For theoretical justification of "self-consistency is not necessarily aligned as an objective with performing better", could the author explain why would policy-value inconsistency not hurt MCTS performance, given that MCTS explicitly uses both value estimates and visit counts for action selection? Could the authors provide a formal characterization of when "simple policy but complex value function" occurs? Under what conditions would this favor ES over gradient-based methods?

---

> ### Author Response · Authors · 2025-12-03
>
> > It would be better put mathematical expressions separately for clear elaboration instead of using in-line mode.
>
> Please state which specific expressions you are referring to. Space constraints make it necessary to inline some expressions.
>
> > for each environment, all contents are put in huge bulk of single paragraph, making it difficult to follow.
>
> We have split these into paragraphs. Thank you for the suggestion.
>
> > The paper lacks theoretical justification. In Section 6 (Discussion), the claim about "simple optimal policy but complex value function" lacks rigor and doesn't explain why this would systematically favor ES.
>
> There are many important papers in machine learning that study phenomena empirically without proving a definitive explanation. In this field, empirical results often outpace theory. ICLR has many papers based on these rather than theoretical analysis.
>
> Our hypothesis in Section 6 *does* explain why AZES outperforms AZ: AZES *directly* optimizes the episode score, whereas AZ does so only indirectly, by trying to minimize a planning loss. AZ crucially relies on learning a good value function. Bad value functions can cause systematically misleading recommendations by MCTS. Our experiments show that AZES, empirically, is more robust to inaccurate value functions.
>
> We have edited the conclusion to explain this more clearly. We have also added new sections to the appendix showing simple, concrete examples where AZ fails while AZES succeeds.
>
> > The paper claims to test "across a wide range of learning rates for fair comparison" between AlphaZero and AlphaZeroES. However, this doesn't ensure a fair comparison. Different learning rates are optimal for different objectives, making it impossible to isolate whether improvements come from the objective change or better hyperparameter selection.
>
> That is why we tested an entire range of learning rates for each method, and compared the **best-performing** ones. This is clearly stated in lines 936-937.
>
> > The loss report is inconsistent. Figures show AlphaZeroES doesn't minimize value/policy losses, yet performance improves. This disconnect isn't adequately explained.
>
> We explain our hypothesis for that in lines 435-448. We have also added new sections to the appendix explaining this phenomenon further.
>
> > Did the authors perform a grid search or other systematic hyperparameter optimization for both methods? If so, what was the protocol?
>
> For a direct apples-to-apples comparison, the pairwise hyperparameter of interest is the learning rate, since AZ and AZES optimize different objectives but are otherwise identical. Therefore, we systematically test a range of learning rates for each of AZ and AZES.
>
> Furthermore, to ensure the superior performance is not purely an artefact of a specific problem size, network size, or network architecture, we include an extensive comparison of these in the appendix.
>
> > How did the authors ensure the learning rate ranges tested were appropriate for each method's objective? Did the authors verify convergence for both methods at their optimal learning rates?
>
> We tested a range of learning rates on a logarithmic scale for each method, as shown in the plot legends. For the statistical tests (Appendix B), we compare each method under their optimal learning rates.
>
> > The ES-specific hyperparameter selection is not explained.
> > How did the authors select the perturbation scale of 0.1 for ES?
>
> This is the most common scale we found in the literature, as well as in OpenAI’s own blog post presenting OpenAI ES.
>
> > Sensitivity analysis is not provided regarding perturbation scale, which is set to be 0.1.
> > What happens with different scales, such as 0.01, 0.05, 0.2, or adaptive schedules?
>
> We have added yet another hyperparameter sweep for the ES scale. It can be found in the appendix.
>
> However, the fact that AZES outperforms AZ at a particular ES scale is already sufficient. Maximizing over more ES scales can only further improve the performance of AZES.
>
> > For theoretical justification of "self-consistency is not necessarily aligned as an objective with performing better", could the author explain why would policy-value inconsistency not hurt MCTS performance, given that MCTS explicitly uses both value estimates and visit counts for action selection? Could the authors provide a formal characterization of when "simple policy but complex value function" occurs?
>
> We have edited the conclusion to go into further detail on this issue. We have also added new sections to the appendix showing simple, concrete examples where AZ fails while AZES succeeds.
>
> > Under what conditions would this favor ES over gradient-based methods?
>
> The superiority of AZES over AZ is due not to a superiority of ES over gradient-based methods, but rather the fact that AZES *optimizes a different objective*, namely, it directly maximizes the episode score. The ES is merely a means to this end. This is explained in lines 48-50 and 186-188.

---

### Official Review · Reviewer_F2RQ · 2025-10-30

**Soundness:** 2
**Presentation:** 3
**Contribution:** 2
**Rating:** 4
**Confidence:** 4

**Summary:**

The authors demonstrate that one can use evolution strategies with an AlphaZero setup to directly optimize maximizing the reward. This appears to work well--or better than the base version in the tested games.

**Strengths:**

* It is always interesting to see ES
* The authors test their method on a number of games
* The authors provide and extensive survey of work in the area in the appendix
* The authors highlighted a few contexts where the ES strategy also did a good job of optimizing the auxiliary tasks that AZ targets.

**Weaknesses:**

* Are there any other baselines or methods (or families of methods AC) you could provide? Here, you only show AZ.
* Overall, (1) it is unclear what the score lines mean, what is a good score / success, (2) relatedly, if AZ is not learning at all or having any success.

**Questions:**

The main question I still have is a want for more context: What type of baselines could you provide, what kinds of other RL agents are typically used for these tasks and how would they compare? Were the AZ agents fairly tested?

* Are the results good? You outperform AZ but it is not clear from the paper would a good performance is. Do other models do much better on these tasks? If AZ models do very poorly and perhaps are under-trained, then doing better is not necessarily super meaningful. Adding context of what success looks like for each task would help.
* Could you try other ways of organizing the plots? I don't think you need to show the loss/value for everything. Make the primary score plots bigger and more readable?
* How much compute is done per model/run ("4 hours of training time per trial") vs what is needed? Did the model training converge?
* What are the standard approaches for these tasks? Is this within the scope under which AZ works? I do see the appendix figures where ES still always did better even under increased compute.

# Minor notes
* L289 - Seem to "increase"

---

> ### Author Response · Authors · 2025-12-03
>
> > Are there any other baselines or methods (or families of methods AC) you could provide? Here, you only show AZ.
>
> Only AZ is necessary. As stated in lines 46-48:
>
> > In this paper, we set out to explore whether we can outperform AlphaZero and its variants in single-agent environments by directly maximizing the episode score instead, while **leaving all other aspects of the agent, MCTS algorithm, and neural architecture unchanged**.
>
> and lines 162-164:
>
> > For clarity, we emphasize that we use the **exact same architecture for both AlphaZero and AlphaZeroES in each problem. This is an apples-to-apples comparison.** The only thing that changes is the optimization objective.
>
> Our goal is to do a direct, pairwise, apples-to-apples comparison with AZ (which is widely used). Other methods are irrelevant.
>
> > (1) it is unclear what the score lines mean, what is a good score / success
>
> We explain what the score means in each environment’s individual description.
>
> > (2) relatedly, if AZ is not learning at all or having any success.
>
> AZ is indeed learning in some environments, for some hyperparameters. This is shown, for example, in Figures 2, 4, and 12. However, in some cases, it struggles to learn.
>
> > What type of baselines could you provide, what kinds of other RL agents are typically used for these tasks and how would they compare?
>
> Lines 93-107 show, with references, that MCTS is typically used for these tasks. The most prominent MCTS-based approach is AZ, which is also state-of-the-art.
>
> > Were the AZ agents fairly tested?
>
> Yes. We test a range of learning rates for both AZ and AZES. We also include extensive hyperparameter comparisons for problem size, network size, and architecture in the appendix.
>
> > Could you try other ways of organizing the plots? I don't think you need to show the loss/value for everything. Make the primary score plots bigger and more readable?
>
> Done. Thank you for the suggestion.
>
> > How much compute is done per model/run ("4 hours of training time per trial")
>
> As stated in lines 241-242:
>
> > We used an NVIDIA A100 SXM4 40GB GPU. Each trial uses 1 such GPU all to itself. This keeps the comparison between AlphaZero and AlphaZeroES as precise as possible.
>
> > vs what is needed?
>
> It is unclear what you mean by “what is needed”.
>
> > Did the model training converge?
>
> For some hyperparameters, yes. For others, no. This is shown in each individual plot.
>
> > What are the standard approaches for these tasks? Is this within the scope under which AZ works?
>
> As explained above and in lines 93-107 of the paper, AZ is state-of-the-art for these types of tasks.

---

### Official Review · Reviewer_f9Th · 2025-11-01

**Soundness:** 1
**Presentation:** 2
**Contribution:** 2
**Rating:** 2
**Confidence:** 4

**Summary:**

The paper focuses on enhancing AlphaZero algorithm with evolution-based optimization algorithms, which allows to directly optimize cumulative reward function over an episode. The core idea is to train actor model with ES algorithm rather then gradient descent.

**Strengths:**

- One of the huge advantages of this method is it ability to scale to a large number of workers, which allows to train agent in parallel setting.

**Weaknesses:**

- No direct comparison to AlphaZero and other methods. Experiments only show an effect of different hyperparameters on AlphaZeroES performance.
- The description of resulting algorithm is hard to understand from pure text description. There is no outline (i.e. step-by-step description or pseudocode) provided in the main part of the paper.

**Questions:**

- Is there any direct comparison with AlphaZero and other RL methods in terms of performance?
- The main motivation of applying ES is direct black-box optimization of cumulative reward. However, RL algorithms such as REINFORCE can do that to. Even more, the agent learning algorithm of original AlphaZero can do that too via n-step returns or lambda-returns. Is there any other motivation for applying ES specifically?

---

> ### Author Response · Authors · 2025-11-14
>
> Dear Reviewer f9Th,
>
> Thank you very much for your feedback.
>
> > No direct comparison to AlphaZero and other methods. Experiments only show an effect of different hyperparameters on AlphaZeroES performance.
>
> Every single plot is a direct comparison to AlphaZero. In each plot legend, the label "es = 0" corresponds to AlphaZero. This is stated in line 246.
>
> > The description of resulting algorithm is hard to understand from pure text description. There is no outline (i.e. step-by-step description or pseudocode) provided in the main part of the paper.
>
> A step-by-step description is given in lines 207-227. Pseudo-code is given in Algorithm 2 of Salimans et al. 2017, which is cited immediately before that. We can paste that pseudo-code here, with attribution, if you prefer that.
>
> We also provide full code in Appendix F.
>
> > The main motivation of applying ES is direct black-box optimization of cumulative reward. However, RL algorithms such as REINFORCE can do that to. Even more, the agent learning algorithm of original AlphaZero can do that too via n-step returns or lambda-returns. Is there any other motivation for applying ES specifically?
>
> REINFORCE requires differentiating the log probabilities of sampled actions. Since the log probabilities are output by MCTS, this requires differentiating through MCTS. But MCTS is not differentiable. This is explained in lines 48-50:
>
> > Since **MCTS is not differentiable**, to maximize the episode score, we employ evolution strategies, a family of algorithms for zeroth-order black-box optimization.
>
> as well as lines 181-188:
>
> > One way to directly optimize the episode score is to use policy gradient methods, which yield an estimator of the gradient of the expected return with respect to the agent’s parameters. There is a vast literature on **policy gradient methods, which include REINFORCE** (Williams, 1992) and actor-critic methods (Konda and Tsitsiklis, 1999; Grondman et al., 2012). Most of these methods assume that the policy is *differentiable*—more precisely, that its output action distribution is differentiable with respect to the parameters of the policy. However, our planning policy uses MCTS as a subroutine, and standard **MCTS is not differentiable**. Because **our policy contains a non-differentiable submodule**, we need to find an alternative way to optimize the policy’s parameters.
>
> (Note: AlphaZero does not require differentiating the output of MCTS, because it only uses that output as a *target* and instead optimizes a different objective, as described in lines 170-176.)
>
> ---
>
> Does this address all of the objections that you raised in your review?

---

### Note · Authors · 2026-01-20

I have read and agree with the venue's withdrawal policy on behalf of myself and my co-authors.